

# Exponentially long lifetime of universal quasi-steady states in topological Floquet pumps

**Tobias Gulden[1,2]⋆, Mark S. Rudner[3], Erez Berg[4,5] and Netanel H. Lindner[1]**

**1** Physics Department, Technion, 3200003 Haifa, Israel
**2** IST Austria, 3400 Klosterneuburg, Austria
**3** Niels Bohr International Academy and Center for Quantum Devices,
University of Copenhagen, 2100 Copenhagen, Denmark
**4** Physics Department, University of Chicago, IL 60637, USA
**5** Faculty of Physics, Weizman Institute of Science, 7610001 Rehovot, Israel

⋆ tgulden@ist.ac.at

## Abstract

We investigate a mechanism to transiently stabilize topological phenomena in long-lived quasi-steady states of isolated quantum many-body systems driven at low frequencies. We obtain an analytical bound for the lifetime of the quasi-steady states which is exponentially large in the inverse driving frequency. Within this lifetime, the quasi-steady state is characterized by maximum entropy subject to the constraint of fixed number of particles in the system's Floquet-Bloch bands. In such a state, all the non-universal properties of these bands are washed out, hence only the topological properties persist.


# 1 Introduction

The possibility to dynamically control material properties through a time-periodic driving field has lead to many proposals [1–22] and experiments [23–26] aimed at realizing topological states in periodically driven systems. Topological properties of materials are usually robust and cannot be changed easily using external perturbations. However, Floquet engineering allows for dynamical modification of topological properties using driving fields. These fields are generally weaker than the bare energy scales of the material system, such as the bandgap in the case of insulators.

A major challenge for Floquet engineering in interacting quantum many body systems is their tendency to absorb energy from the drive and generically heat towards featureless high-entropy states [27–29]. In such a state all correlations are trivial, and any topological properties are washed out. Several exceptions to the fate of reaching a featureless state were proposed, such as Floquet integrable or many-body localized systems [30–40].

The featureless high-entropy fate of interacting Floquet systems at long times may not preclude a system from exhibiting topological phenomena at intermediate times. For example, when subjected to high-frequency drives, the heating rates of quantum many body systems are suppressed. In this case, a long prethermal time window can emerge in which the evolution of the system is governed by an effective time-independent Hamiltonian [41–50]. On the other hand, new types of topological phenomena possible only in periodically driven systems have recently been discovered [3, 8, 12, 17, 21, 51–56] which require that the driving frequency is comparable to or smaller than the natural energy scales of the system. We therefore investigate the conditions under which prethermal states can be stabilized over a long time window for systems driven at low frequencies [57]. This includes, but is not limited, to states with topological properties.

We focus our attention on the dynamics of interacting fermions in slowly driven lattices. We consider a system where the bandstructure of the Hamiltonian at any specific time exhibits two sets of bands separated by a large bandgap. In the limit in which the bandgap is larger than the interaction strength, bandwidths of the individual bands, and the driving frequency, we expect the rates of processes in which particles are scattered across the bandgap to be suppressed relative to intraband scattering processes, due to the large energy mismatch involved in the

former. Therefore, the total populations in bands below and above the bandgap are nearly conserved quantities over a long time interval (before interband scattering becomes significant). Due to intraband scattering processes the system quickly attains a high entropy state, subject to the restriction that the initial total band populations are conserved. In Ref. [57], we showed how such a state emerges in a particular one dimensional system. Moreover, we showed that this state carries a universal quasi-steady state current whose value depends only on the topological character of the Floquet bandstructure and the initial populations of bands.

The goal of this paper is to derive a universal and rigorous bound on the interband scattering rates for interacting fermions in slowly driven lattices. The bound holds for a large class of systems in different spatial dimensions, and with different topological properties. The bound implies that the prethermal regime for low-frequency driving is indeed a general phenomenom which can be found across this wide variety of systems. The bound demonstrates that the band populations are indeed almost conserved quantities on a timescale that is exponentially long in the ratio between the minimal instantaneous bandgap and the maximum among the bandwidth, driving frequency, and interaction strength.

Our results allow us to go beyond the one dimensional pump analyzed in Ref. [57] and set the stage for finding new types of topological transport phenomena in periodically driven systems with high energy density. Such extensions include, for example, driving-induced Weyl points in 3 dimensions [58–60]. Furthermore, since the prethermal states obtained in the low driving frequency regime exhibit a homogenous distribution of particles over all momenta within each band, they can also be used to measure the topological character of the Floquet bands (for example, by measuring the average of the Berry curvature to infer the Chern number of the bands [54, 55]).

This paper is structured as follows: In section 2 we motivate the problem with a generic description of the effects of interactions in slowly-driven systems, in particular in topological pumps, and describe the setup of the problem. In the main text we restrict ourselves to a 1-dimensional model with 2 bands and on-site interaction. In appendix E we generalize the calculation to multi-band systems in any number of spatial dimensions, with generic short-range interactions. Here we want to emphasize that our derivation is independent of the topological nature of the system, but applies to all slowly-driven systems with a large bandgap compared to driving frequency, instantaneous bandwidth, and interaction strength. In Section 3 we differentiate between two qualitatively different classes of contributions to the excitation rate which we treat separate thereafter. Crucial for this distinction are locality properties of the operators which we discuss in section 3.2, together with introducing key quantities and notation for our analytical treatment. By using Lieb-Robinson bounds in section 4.1 we show that one of the classes above only yields exponentially small contributions and can be neglected. In section 4.2 we bound the remaining contributions. We show that the density of excitations grows at a rate that is exponentially small in the ratio of the bandgap over the maximum of driving frequency, bandwidth, and interaction strength. In section 5 we discuss our findings and their implications.

## 2 Problem setup and main results

In this section we describe the setup of the problem in this paper and define the basic notations that will be used throughout. Using these definitions, we give a formal statement of the main result which then will be derived in the subsequent sections.

For simplicity, throughout the main text we will focus on spinless fermions hopping on a one-dimensional lattice with two atoms in each unit cell. Our results can be straightforwardly generalized to higher dimensions and larger numbers of bands or degrees of freedom, as dis-

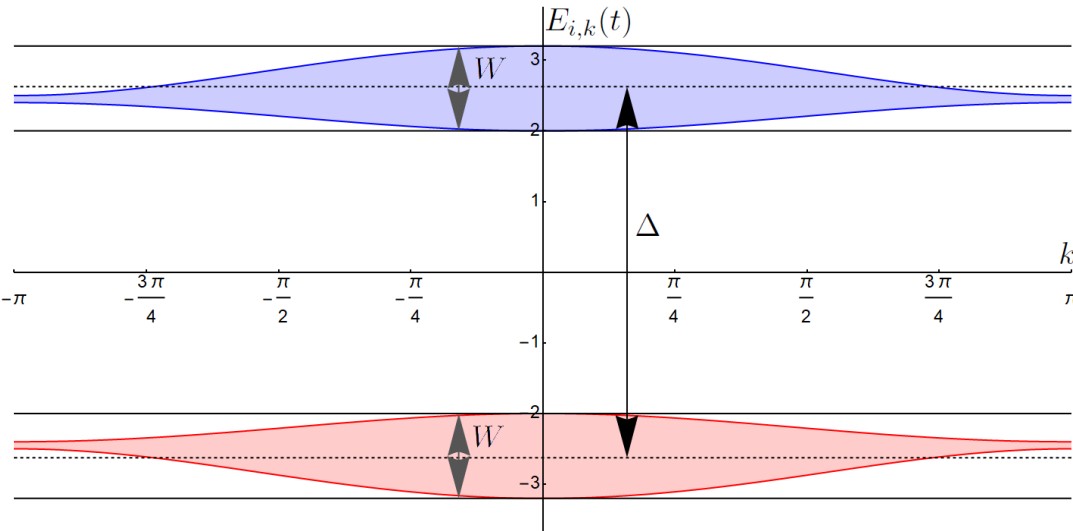

Figure 1: Plot of the instantaneous bands $\tilde{E}_{i,k}(t)$ of a periodically driven one-dimensional lattice system. Details of the used model are in appendix A, which is an example of a topological pump. The solid lines are the minimum and maximum over all times $t$ for each momentum $k$, the shaded areas are the possible instantaneous eigenenergies. We define the energy scale $W$ as the width of the bands, while $\Delta$ denotes the separation of the centers of the bands. For $\Delta > W$ the system is gapped at all times $t$. In the following we refer to $\Delta$ as the bandgap.

cussed in appendix E. The dynamics of fermions on the lattice is described by a time-periodic Bloch Hamiltonian

$$\tilde{\mathcal{H}}_0(t) = \sum_k \left(A_k^\dagger \, B_k^\dagger\right) \begin{pmatrix} h_{\mathrm{AA},k}(t) & h_{\mathrm{AB},k}(t) \\ h_{\mathrm{BA},k}(t) & h_{\mathrm{BB},k}(t) \end{pmatrix} \begin{pmatrix} A_k \\ B_k \end{pmatrix}, \tag{1}$$

where $A_k^\dagger$ and $B_k^\dagger$ create Bloch states with crystal momentum $k$ on sublattices $A$ and $B$, respectively, and $h_{\mathrm{AA},k}(t),\ldots,$ are periodic functions of time with a period $T$. A specific lattice Hamiltonian realizing Thouless' one-dimensional pump, which takes the form of Eq. (1), is given in appendix A. For other models realizing topological pumps, see [51, 53–55, 58–65]. The Hamiltonian $\tilde{\mathcal{H}}_0(t)$ yields two instantaneous energy bands $\tilde{E}_{i,k}(t), i = 1, 2$. We will consider the case where the parameters appearing in $\tilde{\mathcal{H}}_0(t)$ are such that its two instantaneous energy bands are separated by a gap at any time $t$. We define the maximal bandwidth of the bands as $W = \max_i \left[\max_{k,t} \tilde{E}_{i,k}(t) - \min_{k,t} \tilde{E}_{i,k}(t)\right]$, and the distance between the bands as $\Delta = \overline{E}_2 - \overline{E}_1$ with the middle of the bands $\overline{E}_i = [\max_{k,t} \tilde{E}_{i,k}(t) + \min_{k,t} \tilde{E}_{i,k}(t)]/2$. In the following we consider the case $\Delta > W$ for which the system is gapped at all times, and refer to $\Delta$ as the bandgap. These definitions are illustrated in Fig. 1.

Using the time-periodicity of the Hamiltonian in Eq. (1) we find a complete basis of single-particle Floquet states $|\tilde{\psi}_{\lambda,k}(t)\rangle = e^{-i\epsilon_{\lambda,k}T}|\tilde{\psi}_{\lambda,k}(t+T)\rangle$ which solve the Schrödinger equation $i\partial_t|\tilde{\psi}\rangle = \tilde{\mathcal{H}}_0(t)|\tilde{\psi}\rangle$. Here $\lambda = R, L$ denotes the two Floquet bands (the choice for the names $R$ and $L$ will be clarified below), and $\epsilon_{\lambda,k}$ are the quasi-energies of the Floquet states. When $\tilde{\mathcal{H}}_0(t)$ realizes a topological pump, the quasi-energies wind around the Floquet-Brillouin zone in the adiabatic limit (see section 2.2 below for more details). Such a spectrum is visualized in Fig. 2. We therefore use the symbol $\tilde{R}_k^\dagger(t)$ and $\tilde{L}_k^\dagger(t)$ to denote creation operators for fermions in the states $|\tilde{\psi}_{R,k}(t)\rangle$ which corresponds to a "right mover" with positive winding, and $|\tilde{\psi}_{L,k}(t)\rangle$ which corresponds to a "left mover" with negative winding. There are actually small gaps from avoided crossings between the right- and left-moving bands in Fig. 2. How-

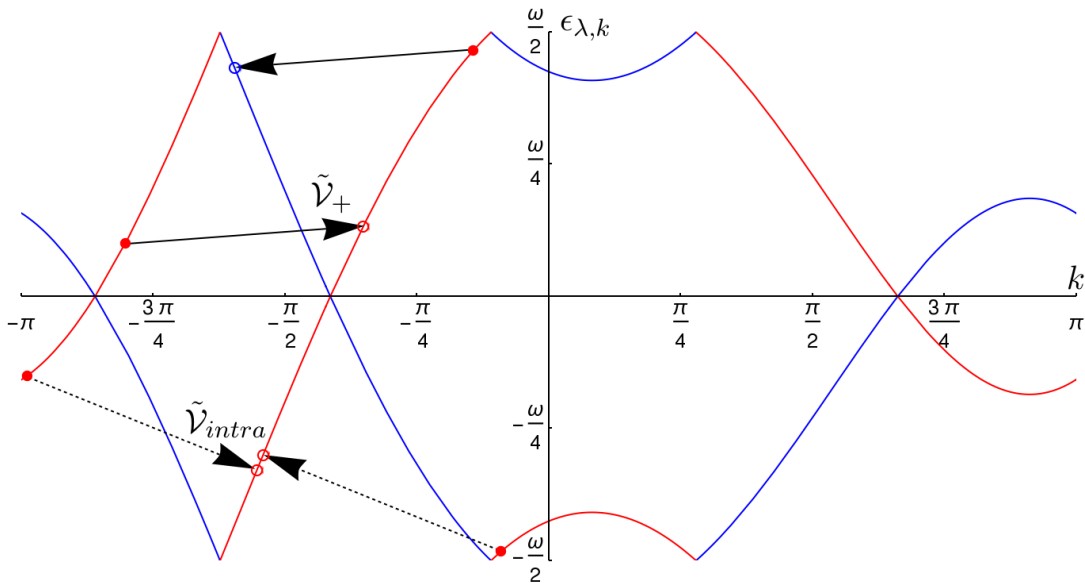

Figure 2: Quasienergies for a topological pump with two bands, which are defined modulo $\omega$. For details of the model see appendix A. The driving frequency $\omega$ is smaller than the intrinsic energy scales. This introduces jumps at quasienergy $\epsilon = \pm\frac{\omega}{2}$. Following the bands throughout the Brillouin zone the periodicity condition gives $\epsilon_{\lambda,k+2\pi} = \epsilon_{\lambda,k} \pm \omega$, which makes the red band a right-moving $R$-band and the blue band a left-moving $L$-band. The arrows denote two different possible processes, one intraband scattering process between two right-moving particles ($\tilde{\mathcal{V}}_{intra}$) and one process where two right-moving particles are scattered such that one particle remains a right-mover while the other is turned into a left-mover ($\tilde{\mathcal{V}}_+$), respectively.

ever these gaps are exponentially small in $\Delta/\omega$ (cf. [57]), hence they can be ignored over exponentially long time scales. We also define the populations $N_R(t) = \sum_k \tilde{R}_k^\dagger(t)\tilde{R}_k(t)$ and $N_L(t) = \sum_k \tilde{L}_k^\dagger(t)\tilde{L}_k(t)$. Importantly, we will show that the band populations $N_R(t)$ and $N_L(t)$ are almost conserved quantities in the prethermal time window that we define below.

## 2.1 Interaction induced quasi-steady states

We study thermalization of the system in the presence of interactions. For concreteness, we consider a two-particle interaction which acts within one unit cell,

$$\mathcal{V} = U\sum_x A_x^\dagger A_x B_x^\dagger B_x. \tag{2}$$

In appendix E we generalize the interaction to any short-range operator. Our results apply for both repulsive and attractive interactions. In the formulas below we assume $U > 0$; for $U < 0$ one should replace $U \to |U|$ throughout.

Generically such a system will eventually reach a featureless infinite-temperature state where all single-particle Floquet eigenstates are populated equally, implying $N_R = N_L$ [27]. Here we study the dynamics at intermediate times before reaching the final state, and prove that there exists a long-lived prethermal state in which $N_R(t)$ and $N_L(t)$ remain close to their initial values.

To bound the rate of relaxation of $N_R(t)$ and $N_L(t)$ we split the interaction term (2) into five different parts, dependent on the change of the number of particles in the two Floquet bands:

$$\mathcal{V} = \tilde{\mathcal{V}}_{intra}(t) + \tilde{\mathcal{V}}_+(t) + \tilde{\mathcal{V}}_{++}(t) + \tilde{\mathcal{V}}_-(t) + \tilde{\mathcal{V}}_{--}(t). \tag{3}$$

The intraband part $\tilde{\mathcal{V}}_{intra}$ preserves the number of particles in each Floquet band, while the interband scattering parts $\tilde{\mathcal{V}}_+$ ($\tilde{\mathcal{V}}_{++}$) and $\tilde{\mathcal{V}}_-$ ($\tilde{\mathcal{V}}_{--}$) transfer one (two) particles from the $R$- to the $L$-band and vice versa, respectively. We make this separation by expressing the interaction $\mathcal{V}$ in Eq. (2) in terms of the $L$- and $R$-band creation and annihilation operators, $\{\tilde{L}_k(t)^\dagger, \tilde{L}_k(t), \tilde{R}_k(t)^\dagger, \tilde{R}_k(t)\}$. The operators $\tilde{\mathcal{V}}_{intra}(t)$, $\tilde{\mathcal{V}}_\pm(t)$, $\tilde{\mathcal{V}}_{++}(t)$ and $\tilde{\mathcal{V}}_{--}(t)$ are thus time-dependent, due to the time-dependence of the Floquet states encoded in $\{\tilde{L}_k(t)^\dagger, \tilde{L}_k(t), \tilde{R}_k(t)^\dagger, \tilde{R}_k(t)\}$.

Consider a system in which the interaction consists only of the band conserving term $\tilde{\mathcal{V}}_{intra}(t)$. Generically, due to the time dependence of the Hamiltonian, after a short thermalization time $\tau_{intra}$ the system reaches a maximal-entropy state subject to the constraint that the band populations $N_R(t)$ and $N_L(t)$ are conserved. In such a state, all momentum states within each band are equally populated.

We focus on times which are longer than the intraband thermalization time $\tau_{intra}$, when the fermions within each band have been fully thermalized. The *many-body Floquet eigenstates* $|\tilde{\Psi}(t)\rangle$ of the Hamiltonian with the band-conserving part of the interaction $\tilde{\mathcal{H}}_0(t) + \tilde{\mathcal{V}}_{intra}(t)$ are all maximal entropy states indexed by its conserved quantities $N_R$ and $N_L$. We will represent prethermal states established after time $\tau_{intra}$ by any one of these states with the appropriate particle numbers.

When the interband scattering parts $\tilde{\mathcal{V}}_\pm$ are included, $N_R(t)$ and $N_L(t)$ are not strictly conserved. To explore the decay of the quasi-steady prethermal state, we consider an initial state with $N_{R,i}$ and $N_{L,i}$ fermions in the $R$- and $L$-band, respectively. We calculate the rate of transferring particles from the $R$- to the $L$-band within Fermi's golden rule,

$$\Gamma(\mathcal{T}) = \frac{1}{L\mathcal{T}} \sum_f \left| \int_0^{\mathcal{T}} dt \langle \tilde{\Psi}_f(t)|\tilde{\mathcal{V}}_+(t)|\tilde{\Psi}_i(t)\rangle \right|^2 . \tag{4}$$

Here $\Gamma(\mathcal{T})$ captures the rate of change of the densities of particles in the $R$- and $L$-band, given by $N_{R,i}/L$ and $N_{L,i}/L$ where $L$ is the length of the system, and $\mathcal{T}$ is much larger than the driving period, $\mathcal{T} \gg T$. The state $|\tilde{\Psi}_i(t)\rangle$ in Eq. (4) is a many-body Floquet eigenstate of $\tilde{\mathcal{H}}_0(t) + \tilde{\mathcal{V}}_{intra}(t)$, with $N_{R,i}$ ($N_{L,i}$) fermions in the $R$- ($L$-)band. Likewise, we take the state $|\tilde{\Psi}_f(t)\rangle$ in Eq. (4) to be a many-body Floquet eigenstate of $\tilde{\mathcal{H}}_0(t) + \tilde{\mathcal{V}}_{intra}(t)$ with $N_{R,i} - 1$ ($N_{L,i} + 1$) particles in the $R$- ($L$-)band (assuming that $N_R > N_L$). The term with the operator operator $\mathcal{V}_{++}(t)$ in Fermi's golden rule also transfers particles from the $R$- to the $L$-band, however as we argue below the rates of such processes are exponentially suppressed compared to the rate in Eq. (4).

Our goal in this work is to bound the rate of transfer of particles from the $R$- to the $L$-band, $\tau_{inter}^{-1} = \lim_{\mathcal{T} \to \infty} \Gamma(\mathcal{T})$. In section 4.2 we prove that interband transitions are strongly suppressed for $\Delta \gg \max(\omega, W, U)$, where $\omega = 2\pi/T$ is the driving angular frequency. Specifically, we show for a general system with short-range interactions in any dimension that the rate of interband transitions is exponentially suppressed,

$$\Gamma \sim \exp\left( -\mu \frac{\Delta}{\max(\omega, W, U)} \right), \tag{5}$$

for a positive constant $\mu$. We note that the process corresponding to the operator $\tilde{\mathcal{V}}_{++}(t)$, in which two particles must traverse the gap $\Delta$ has a contribution to the rate which scales as $\exp\left( -2\mu \frac{\Delta}{\max(\omega, W, U)} \right)$ and is therefore negligible in comparison with $\Gamma(\mathcal{T})$ in Eq. (5). This suppression of interband transitions creates a time window $\tau_{intra} \ll t \ll \tau_{inter}$ during which the particles within the two bands separately are thermalized, while the densities of particles in the $R$- and $L$-bands $n_{R,L} = \frac{N_{R,L}}{L}$ remain conserved quantities up to exponentially small corrections. We refer to this state at intermediate times as a prethermal quasi-steady state.

The combination of maximal entropy and almost conserved quantities in the quasi-steady state leads to the emergence of new universal phenomena. In particular, the homogeneous distribution of particles in the momentum states within each band allows the topological features of the Floquet bandstructure to be manifested in the quasisteady state.

## 2.2 Universality in the quasi-steady state

In the main text we consider a generic one-dimensional case with two bands, for which the quadratic part of the Hamiltonian is given by Eq. (1). The Floquet bandstructure is obtained by examining the spectrum of the unitary matrix $U_T = \mathcal{P}e^{-i\int_0^T dt\, \hat{h}_{0,k}(t)}$, where $\hat{h}_{0,k}(t)$ is the matrix appearing in Eq. (1) and $\mathcal{P}$ refers to path ordering. In the limit $\omega/\Delta \ll 1$ the two Floquet bands may exhibit non-trivial *chirality*. Each quasienergy band $\epsilon_{\lambda,k}$ of $U_T$ must be periodic in $k$. Since the quasienergies are defined in a Floquet-Brillouin zone which is periodic in both $k$ and $\epsilon$, the quasienergy bands can wind an integer number of times in the quasienergy direction when $k$ goes from 0 to $2\pi$ (we take the lattice constant to be 1). This winding is captured by the integer winding number $\nu = \frac{T}{2\pi}\int_0^{2\pi} dk \frac{\partial \epsilon}{\partial k}$. Bands with non-zero $\nu$ are chiral and exhibit a *quantized* non-zero average group velocity $\nu/T$. Note that the sum of the winding numbers for all the bands must necessarily vanish; for two bands, the winding numbers are equal in magnitude and opposite in sign. For any non-zero value of $\omega$, avoided crossings in the Floquet spectrum open at the intersections between counter-propagating Floquet bands. As a result, when these avoided crossings are resolved, all Floquet bands have zero winding number [57, 66]. However, in the limit $\omega/\Delta \ll 1$, the corresponding gaps at these avoided crossings are exponentially suppressed in $\Delta/\omega$. Therefore, chiral Floquet bands with non-zero values of $\nu$ can be defined in the limit $\omega/\Delta \ll 1$, in which the avoided crossings can be ignored, see Fig. 2. The winding number $\nu$ is in fact the topological invariant characterizing Thouless' quantized charge pump [3, 53, 57].

For chiral Floquet bands, the system may carry a nonvanishing current in the quasi-steady state. Recall that in the quasi-steady state the distribution of particles is uniform as a function of momentum within each band. Therefore, we expect the contribution to the current carried by each band to be given by the average group velocity of the band times the density of particles $n_{R,L}$ occupying this band [57] (setting $e = 1$ for the charge of the particles). Using the quantization of the averaged group velocity described above, this leads to a simple form for the quasi-steady state current:

$$j = (n_R - n_L)\frac{\nu}{T}. \tag{6}$$

Remarkably, this result is universal and does not depend on the details of the bandstructure. The mechanism we describe here extends beyond the one dimensional example discussed above. In fact, non-trivial topology of the Floquet bands in a variety of slowly-driven systems can similarly be reflected in other universal quasi-steady state observables. In appendix E we generalize our results to systems in any dimension with an arbitrary number of bands to prove the general existence of quasi-steady states.

# 3 Locality structure of the contributions to the excitation rate

In this section we show how to separate the contributions to the excitation rate in Eq. (4) into two types of terms which we call *nearby* and *distant* terms. We define this decomposition in section 3.1. In section 3.2 we explicitly show that the intra- and interband interaction operators, $\tilde{\mathcal{V}}_{intra}$, $\tilde{\mathcal{V}}_{\pm}$, and $\tilde{\mathcal{V}}_{\pm\pm}$ in Eq. (3), can each be written as a sum of quasi-local terms (see definition below), which is crucial for this separation. The notations defined in this section will be used throughout the rest of this paper.

### 3.1 Decomposition into nearby and distant terms

The interaction in Eq. (2) is a sum over local terms $A_x^\dagger A_x B_x^\dagger B_x$. We rewrite each term in the basis of the Floquet eigenstate operators $\tilde{L}_k^\dagger, \tilde{R}_k^\dagger$. In this basis we can distinguish between terms that conserve the population numbers $N_{R,L}$, which make up $\tilde{\mathcal{V}}_{intra}$, and terms which move particles from the $R$- to the $L$- band forming $\tilde{\mathcal{V}}_+$ and $\tilde{\mathcal{V}}_{++}$ (or vice versa for $\tilde{\mathcal{V}}_-$ and $\tilde{\mathcal{V}}_{--}$). For the interband interaction operator which moves one particle we write

$$\tilde{\mathcal{V}}_+(t) = \sum_x \tilde{\mathcal{V}}_x(t), \tag{7}$$

where each $\tilde{\mathcal{V}}_x(t)$ is the corresponding component of $A_x^\dagger A_x B_x^\dagger B_x$. The square of the matrix element in Eq. (4) produces a double-sum over $x$ and $x'$ and double-integral over $t$ and $t'$,

$$\Gamma = \frac{1}{L\mathcal{T}} \sum_f \iint_0^{\mathcal{T}} dt' dt \sum_{x',x=1}^{L} \langle \tilde{\Psi}_i(t') | \tilde{\mathcal{V}}_{x'}^\dagger(t') | \tilde{\Psi}_f(t') \rangle \langle \tilde{\Psi}_f(t) | \tilde{\mathcal{V}}_x(t) | \tilde{\Psi}_i(t) \rangle. \tag{8}$$

To separate the terms by distance we define a fixed radius $r^*$ according to which we split thedouble-sum into $\sum_{x',x=1}^{L} = \sum_{x=1}^{L} \sum_{|x'-x|>r^*} + \sum_{x=1}^{L} \sum_{|x'-x| \le r^*}$. We refer to the first part as *distant* and the second part as *nearby* terms. However, note that due to the projection on the Floquet bands the operators $\tilde{\mathcal{V}}_x(t)$ are not strictly local. For this separation to be meaningful we first need to show that $\tilde{\mathcal{V}}_x(t)$ is *quasi-local*. To define quasi-locality around the site $x$ we express an operator $\tilde{\mathcal{O}}_x$ in terms of annihilation and creation operators in the site basis, i.e., the operators $A_{x'}^\dagger, B_{x'}^\dagger, A_{x'}, B_{x'}$. We call the operator $\tilde{\mathcal{O}}_x$ quasi-local around $x$ if the coefficients in this expansion decay at least exponentially with the distance $r = |x-x'|$ of any contributing creation/annihilation operator from the site $x$.

### 3.2 Local representation of the intra- and interband interaction operators

The creation operators for the single-particle Floquet-Bloch states are related to those in the sublattice basis via a unitary transformation, which in general can be written as

$$\begin{pmatrix} \tilde{R}_k^\dagger(t) \\ \tilde{L}_k^\dagger(t) \end{pmatrix} = \begin{pmatrix} \tilde{\alpha}_k(t) & \tilde{\beta}_k(t) \\ -\tilde{\beta}_k(t)^* e^{-i\tilde{\theta}_k(t)} & \tilde{\alpha}_k(t)^* e^{-i\tilde{\theta}_k(t)} \end{pmatrix} \begin{pmatrix} A_k^\dagger \\ B_k^\dagger \end{pmatrix}. \tag{9}$$

Note that the periodicity condition of the Floquet-Bloch states carries over to the basis transformation functions, $\tilde{\alpha}_{k+2\pi}(t) = \tilde{\alpha}_k(t) = e^{i\epsilon_{R,k}T} \tilde{\alpha}_k(t+T)$, and similarly for $\tilde{\beta}_k(t)$. With this definition we can write the explicit form of the interband interaction operator $\tilde{\mathcal{V}}_+$ in Eq. (7),

$$\tilde{\mathcal{V}}_+(t) = \sum_x \tilde{\mathcal{V}}_x(t) = \sum_x \left( \tilde{\mathcal{V}}_x^{RR \to RL}(t) + \tilde{\mathcal{V}}_x^{RL \to LL}(t) \right), \tag{10}$$

where

$$\tilde{\mathcal{V}}_x^{RR \to RL}(t) = \frac{U}{L^2} \sum_{\{k_i\}} e^{ix(k_1+k_2-k_3-k_4)} \tilde{f}_R(\{k_i\}, t) \tilde{L}_{k_1}^\dagger(t) \tilde{R}_{k_2}^\dagger(t) \tilde{R}_{k_3}(t) \tilde{R}_{k_4}(t), \tag{11}$$

$$\tilde{\mathcal{V}}_x^{RL \to LL}(t) = \frac{U}{L^2} \sum_{\{k_i\}} e^{ix(k_1+k_2-k_3-k_4)} \tilde{f}_L(\{k_i\}, t) \tilde{L}_{k_1}^\dagger(t) \tilde{L}_{k_2}^\dagger(t) \tilde{L}_{k_3}(t) \tilde{R}_{k_4}(t),$$

with the two functions $\tilde{f}_R(\{k_i\}, t) = \left( \tilde{\alpha}_{k_1}(t) \tilde{\alpha}_{k_2}^*(t) + \tilde{\beta}_{k_1}(t) \tilde{\beta}_{k_2}^*(t) \right) \tilde{\alpha}_{k_3}(t) \tilde{\beta}_{k_4}(t) e^{i\tilde{\theta}_{k_1}(t)}$ and $\tilde{f}_L(\{k_i\}, t) = \tilde{\alpha}_{k_1}(t) \tilde{\beta}_{k_2}(t) e^{i(\tilde{\theta}_{k_1}(t)+\tilde{\theta}_{k_2}(t)-\tilde{\theta}_{k_3}(t))} \left( \tilde{\alpha}_{k_3}(t)^* \tilde{\alpha}_{k_4}(t) + \tilde{\beta}_{k_3}^*(t) \tilde{\beta}_{k_4}(t) \right)$. $\tilde{\mathcal{V}}_x^{RR \to RL}(t)$ represents the scattering process of two particles from the $R$-band to one $R$- and one $L$-particle.

$\tilde{\mathcal{V}}_x^{\mathrm{RL}\to\mathrm{LL}}(t)$ describes a scattering process of two particles from opposite bands when both end up in the $L$-band. Note that the operator $\tilde{\mathcal{V}}_+$ is not Hermitian, its adjoint operator is $\tilde{\mathcal{V}}_-$.

The functions $\tilde{\alpha}_k, \tilde{\beta}_k$ are periodic in $k$, and in general their Fourier series only contains low modes in $k$. From this it follows that at all times $\tilde{\mathcal{V}}_x(t)$ is quasi-local around $x$, as defined above. Hence we classify each term in Eq. (8), which involves operators $\tilde{\mathcal{V}}_x(t)$ and $\tilde{\mathcal{V}}_{x'}(t')$, as "nearby" if $|x - x'| \le r^*$ or "distant" if $|x - x'| > r^*$.

In section 4.1 we will use Lieb-Robinson bounds to show that the distant terms in Eq. (8) are negligible. Lieb-Robinson bounds can be applied if the Hamiltonian of the system is a sum of quasi-local terms at all times $t$. As mentioned in section 2, time evolution of the states $|\tilde{\Psi}_{i/f}(t)\rangle$ is given by the Hamiltonian $\tilde{\mathcal{H}}(t) = \tilde{\mathcal{H}}_0(t) + \tilde{\mathcal{V}}_{intra}(t)$. In general the quadratic part, Eq. (1), is a sum of local operators. To analyze locality properties of the band population conserving interaction $\tilde{\mathcal{V}}_{intra}$ we rewrite it in the Floquet band basis defined in Eq. (9). This gives a sum of three types of terms,

$$\tilde{\mathcal{V}}_{intra}(t) = \tilde{\mathcal{V}}_{intra}^{\mathrm{RR}}(\{q_i\}, t) + \tilde{\mathcal{V}}_{intra}^{\mathrm{LR}}(\{q_i\}, t) + \tilde{\mathcal{V}}_{intra}^{\mathrm{LL}}(\{q_i\}, t). \tag{12}$$

Here

$$\tilde{\mathcal{V}}_{intra}^{\mathrm{RR}}(t, \{q_i\}) = \frac{U}{L^2} \sum_{x=1}^{L} \sum_{\{q_i\}} e^{ix(q_1+q_2-q_3-q_4)} \tilde{g}_{\mathrm{RR}}(\{q_i\}, t) \tilde{R}_{q_1}^{\dagger}(t) \tilde{R}_{q_2}^{\dagger}(t) \tilde{R}_{q_3}(t) \tilde{R}_{q_4}(t), \tag{13}$$

with the function $\tilde{g}_{\mathrm{RR}}(\{q_i\}, t) = -\tilde{\alpha}_{q_1}^*(t)\tilde{\beta}_{q_2}^*(t)\tilde{\alpha}_{q_3}(t)\tilde{\beta}_{q_4}(t)$, describes scattering of two particles within the $R$-band. The band population conserving interaction between one particle in the $R$-band and one particle in the $L$-band is described by

$$\tilde{\mathcal{V}}_{intra}^{\mathrm{LR}}(\{q_i\}, t) = \frac{U}{L^2} \sum_{x=1}^{L} \sum_{\{q_i\}} e^{ix(q_1+q_2-q_3-q_4)} \tilde{g}_{\mathrm{LR}}(\{q_i\}, t) \tilde{L}_{q_1}^{\dagger}(t) \tilde{R}_{q_2}^{\dagger}(t) \tilde{L}_{q_3}(t) \tilde{R}_{q_4}(t), \tag{14}$$

where

$$\tilde{g}_{\mathrm{LR}}(\{q_i\}, t) = -\left(\tilde{\alpha}_{q_1}(t)\tilde{\alpha}_{q_2}^*(t) + \tilde{\beta}_{q_1}(t)\tilde{\beta}_{q_2}^*(t)\right)\left(\tilde{\alpha}_{q_3}(t)^*\tilde{\alpha}_{q_4}(t) + \tilde{\beta}_{q_3}^*(t)\tilde{\beta}_{q_4}(t)\right)e^{i(\tilde{\theta}_{q_1}(t)-\tilde{\theta}_{q_3}(t))}.$$

The operator $\tilde{\mathcal{V}}_0^{LL}(t, \{q_i\})$ contains only creation and annihilation operators in the $L$-band,

$$\tilde{\mathcal{V}}_{intra}^{\mathrm{LL}}(t, \{q_i\}) = \frac{U}{L^2} \sum_{x=1}^{L} \sum_{\{q_i\}} e^{ix(q_1+q_2-q_3-q_4)} \tilde{g}_{\mathrm{LL}}(\{q_i\}, t) \tilde{L}_{q_1}^{\dagger}(t) \tilde{L}_{q_2}^{\dagger}(t) \tilde{L}_{q_3}(t) \tilde{L}_{q_4}(t), \tag{15}$$

with $\tilde{g}_{\mathrm{LL}}(\{q_i\}, t) = -\tilde{\beta}_{q_1}(t)\tilde{\alpha}_{q_2}(t)\tilde{\beta}_{q_3}^*(t)\tilde{\alpha}_{q_4}^*(t)$. Using similar reasoning to the argument demonstrating quasi-locality of $\tilde{\mathcal{V}}_x$ above, the periodicity of $\tilde{\alpha}_q, \tilde{\beta}_q$ can be used to show that at all times $\tilde{\mathcal{V}}_{intra}(t)$ is a sum of quasi-local terms. This property is crucial for the bounds for both the distant and the nearby terms which we will derive in the next section.

## 4  Derivation of an upper bound on the excitation rate

With these preliminaries we are prepared to derive a rigorous upper bound on the excitation rate $\Gamma$ in Eq. (8). To this end we consider separately the contribution of the distant terms and of the nearby terms, as discussed in section 3.1, and derive an upper bound for each.

### 4.1 Lieb-Robinson bound on the contribution of the distant terms

We begin with the distant terms in Eq. (8), where $|x'-x| > r^*$. Lieb-Robinson bounds [67,68] provide a bound on the expectation value of the commutator of two quasi-local operators, evaluated at different times in the Heisenberg picture. We aim to rewrite Eq. (8) with a commutator. Therefore we subtract the term

$$\langle \tilde{\Psi}_i(t)|\tilde{\mathcal{V}}_x(t)|\tilde{\Psi}_f(t)\rangle \langle \tilde{\Psi}_f(t')|\tilde{\mathcal{V}}_{x'}(t')^\dagger|\tilde{\Psi}_i(t')\rangle = 0 \tag{16}$$

from the argument of the sum over $|x - x'| > r^*$. This term equals zero because the adjoint operator $\tilde{\mathcal{V}}_{x'}(t')^\dagger$ moves a particle from the $L$-band to the $R$-band, however by assumption $|\tilde{\Psi}_f(t')\rangle$ has one particle more in the $L$-band than $|\tilde{\Psi}_i(t')\rangle$. Next we switch to the Heisenberg picture where the states are time-independent and the operators evolve with the time-evolution operator corresponding to the Hamiltonian $\tilde{\mathcal{H}}(t) = \tilde{\mathcal{H}}_0(t) + \tilde{\mathcal{V}}_{intra}(t)$. Then we can sum over all final states, which in the subspace of fixed particle numbers in each band gives $\sum_f |\tilde{\Psi}_f\rangle\langle\tilde{\Psi}_f| = \mathbb{1}$, and rewrite the contribution of the distant terms in Eq. (8) as

$$\Gamma_{dist} = \frac{1}{L\mathcal{T}} \iint_0^{\mathcal{T}} dt'dt \sum_{x=1}^{L} \sum_{|x'-x|>r^*} \langle \tilde{\Psi}_i|[\tilde{\mathcal{V}}_{x'}^{H\dagger}(t'), \tilde{\mathcal{V}}_x^H(t)]|\tilde{\Psi}_i\rangle. \tag{17}$$

Here $\tilde{\mathcal{V}}_x^H(t)$ is the equivalent operator in the Heisenberg picture.

Following the discussion in section 3.2, the Hamiltonian $\tilde{\mathcal{H}}(t) = \tilde{\mathcal{H}}_0(t) + \tilde{\mathcal{V}}_{intra}(t)$ is a sum of quasi-local terms at all times $t$, which is a necessary requirement for the application of the Lieb-Robinson bound. Under this condition, the norm of the commutator of time-dependent quasi-local operators in the Heisenberg picture is bounded by [67,68]

$$\|[\tilde{\mathcal{V}}_{x+r}^{H\dagger}(t'), \tilde{\mathcal{V}}_x^H(t)]\| \leq C\|\tilde{\mathcal{V}}_{x+r}^{H\dagger}\|\|\tilde{\mathcal{V}}_x^H\|e^{-c(|r|-v|t-t'|)}, \tag{18}$$

where the usual operator norm is defined as $\|\tilde{\mathcal{V}}\|^2 \equiv \sup_{|\tilde{\psi}|=1}\langle\tilde{\psi}|\tilde{\mathcal{V}}^\dagger\tilde{\mathcal{V}}|\tilde{\psi}\rangle$. For each quasi-local term the norm of the interaction is bounded by $\|\tilde{\mathcal{V}}_x^H\| \leq U$. In Eq. (18) $C$, $c$, and $v$ are numerical constants, where $v$ is referred to as the Lieb-Robinson velocity. After substituting the bound (18) into Eq. (17) and using translational invariance we are left with a double time integral and a sum over $r = |x' - x|$. Converting the sum over $r$ into an integral (with $a = 1$ for the lattice spacing), we obtain:

$$\begin{aligned} \Gamma_{dist} &\leq \frac{CU^2}{\mathcal{T}} 2 \int_{r^*}^{\infty} dr e^{-cr} \iint_0^{\mathcal{T}} dt'dt\, e^{cv|t-t'|} \\ &= \frac{4C}{c^3v^2} \frac{U^2}{\mathcal{T}} e^{-cr^*} \left(e^{cv\mathcal{T}} - 1 - cv\mathcal{T}\right). \end{aligned} \tag{19}$$

For times $\mathcal{T} < \frac{r^*}{v}$ this contribution is exponentially small. If we pick $r^*$ large enough the contribution of the distant terms is negligible compared to the nearby terms, which we discuss next.

### 4.2 Bounding the nearby terms

We now turn to the nearby terms in Eq. (8),

$$\Gamma_{near} = \frac{1}{L\mathcal{T}} \sum_f \sum_{x=1}^{L} \sum_{|r|\leq r^*} \iint_0^{\mathcal{T}} dt'dt \langle \tilde{\Psi}_i|\tilde{\mathcal{V}}_{x+r}^\dagger(t')|\tilde{\Psi}_f\rangle \langle\tilde{\Psi}_f|\tilde{\mathcal{V}}_x(t)|\tilde{\Psi}_i\rangle. \tag{20}$$

We proceed in two steps. First we move to a rotating frame in which all instantaneous eigenstates of $\tilde{\mathcal{H}}_0$ lie within a narrow energy window of width $W$. Through this transformation

the gap $\Delta$ turns into a high-frequency drive, in addition to the low-frequency drive $\omega$. In a system driven by a high frequency, the excitation rate is strongly suppressed, as shown in Ref. [46]. We use this intuition to derive a bound in the presence of both a high- and a low-frequency drive.

As a first step toward bounding the magnitude of the contribution in Eq. (20), we use the triangle inequality

$$\mathcal{I}_x \mathcal{I}_{x'}^* + \mathcal{I}_{x'} \mathcal{I}_x^* \le |\mathcal{I}_x|^2 + |\mathcal{I}_{x'}|^2, \tag{21}$$

with $\mathcal{I}_x = \int_0^{\mathcal{T}} dt \langle \tilde{\Psi}_f | \tilde{\mathcal{V}}_x(t) | \tilde{\Psi}_i \rangle$, to show that the contributions of mixed terms with different $x$ and $x'$ are bounded by terms with both operators centered at the same site $x$. Hence we can reduce the calculation to same-site terms through the bound

$$\Gamma_{near} \le \frac{2r^* + 1}{L\mathcal{T}} \sum_f \sum_{x=1}^L \left| \int_0^{\mathcal{T}} dt \langle \tilde{\Psi}_f | \tilde{\mathcal{V}}_x(t) | \tilde{\Psi}_i \rangle \right|^2, \tag{22}$$

where $2r^* + 1$ is the number of terms with $|r| \le r^*$.

### 4.2.1 Rotating frame transformation

As discussed in section 2 we assume that at all times $t$ the instantaneous single-particle eigenstates fall into two bands of width bounded by $W$, which are separated by a large bandgap $\Delta \gg W$. For a slowly-driven system with $\omega \ll \Delta$ the Floquet eigenstates and the instantaneous eigenstates are almost identical. Most importantly, the diagonal elements $\tilde{E}_{11,k}(t), \tilde{E}_{22,k}(t)$ of the Hamiltonian in the basis $\{\tilde{L}_k^\dagger(T), \tilde{R}_k^\dagger(t)\}$ and the instantaneous eigenenergies $E_{1,k}(t), E_{2,k}(t)$ agree up to small corrections of order $\mathcal{O}(\omega/\Delta)^2$ [69]. Hence the diagonal elements also fall into a narrow interval, $\max_{k,t} \tilde{E}_{ii,k}(t) - \min_{k,t} \tilde{E}_{ii,k}(t) \le W$ for $i = 1, 2$. We now perform a rotating frame transformation to shift the diagonal elements to the interval $[-W/2, W/2]$, see illustration in Fig. 3. To this end we define new operators as

$$R_k^\dagger(t) = \tilde{R}_k^\dagger(t) e^{-i\Delta t/2} \quad ; \quad L_k^\dagger(t) = \tilde{L}_k^\dagger(t) e^{i\Delta t/2}. \tag{23}$$

These operators define Floquet eigenstates which solve the time-dependent Schrödinger equation for the transformed Hamiltonian

$$
\begin{aligned}
\mathcal{H}_0(t) &= \sum_k \begin{pmatrix} R_k^\dagger & L_k^\dagger \end{pmatrix} \begin{pmatrix} \tilde{E}_{11,k} + \frac{\Delta}{2} & \tilde{E}_{12,k} e^{i\Delta t} \\ \tilde{E}_{21,k} e^{-i\Delta t} & \tilde{E}_{22,k} - \frac{\Delta}{2} \end{pmatrix} \begin{pmatrix} R_k \\ L_k \end{pmatrix} \\
&= \sum_k \begin{pmatrix} R_k^\dagger & L_k^\dagger \end{pmatrix} \begin{pmatrix} E_{11,k} & E_{12,k} \\ E_{21,k} & E_{22,k} \end{pmatrix} \begin{pmatrix} R_k \\ L_k \end{pmatrix}.
\end{aligned}
\tag{24}
$$

The quantities $\tilde{E}_{ij,k}(t)$ are obtained from the original Hamiltonian in Eq. (1) via the change of basis in Eq. (9). As a consequence, for all $k$ and $t$ the absolute values of the new diagonal elements are bounded by half the bandwidth, $|E_{11,k}(t)|, |E_{22,k}(t)| \le \frac{W}{2}$. The same applies to the instantaneous eigenenergies in Fig. 3, $|E_{i,k}(t)| \le \frac{W}{2}$. We define dimensionless quantities $e_{ij,k}(t) \equiv E_{ij,k}(t)/W$ and rewrite the Hamiltonian in Eq. (24) as

$$\mathcal{H}_0(t) = W \sum_k \begin{pmatrix} R_k^\dagger & L_k^\dagger \end{pmatrix} \begin{pmatrix} e_{11,k} & e_{12,k} \\ e_{21,k} & e_{22,k} \end{pmatrix} \begin{pmatrix} R_k \\ L_k \end{pmatrix}. \tag{25}$$

Furthermore the diagonal elements $e_{11,k}(t), e_{22,k}(t)$ of the transformed Hamiltonian have the same periodicity in both $k$ and $t$ as the original matrix elements. When writing the rotated

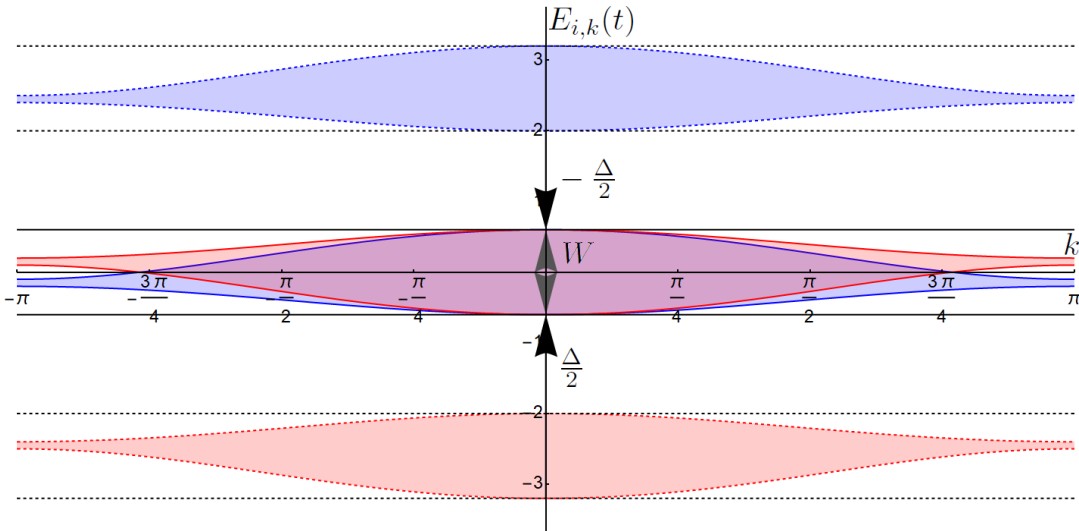

Figure 3: The instantaneous eigenstates of the transformed Hamiltonian (24), for the same model as in Fig. 1 (details of the model are in appendix A). Compared to the original instantaneous eigenstates in Fig. 1 (dashed lines) the rotating frame transformation (23) shifts the bands by $\pm\frac{\Delta}{2}$ so that all rotated eigenstates lie inside a narrow interval of width $W$ (solid lines).

operators in the sublattice basis akin to Eq. (9),

$$
\begin{pmatrix} R_k^\dagger(t) \\ L_k^\dagger(t) \end{pmatrix} = \begin{pmatrix} \alpha_k(t) & \beta_k(t) \\ -\beta_k(t)^* e^{-i\theta_k(t)} & \alpha_k(t)^* e^{-i\theta_k(t)} \end{pmatrix} \begin{pmatrix} A_k^\dagger \\ B_k^\dagger \end{pmatrix}, \tag{26}
$$

the new functions $\alpha_k(t), \beta_k(t)$ differ from the ones defined in Eq. (9) only by a fast oscillating phase,

$$
\alpha_k(t) = \tilde{\alpha}_k(t) e^{-i\Delta t/2} \quad ; \quad \beta_k(t) = \tilde{\beta}_k(t) e^{-i\Delta t/2} \quad ; \quad \theta_k(t) = \tilde{\theta}_k(t). \tag{27}
$$

It turns out that after this transformation the forms of the intraband interaction operator $\mathcal{V}_{intra}$ in Eq. (12) and the interband interaction $\mathcal{V}_+$ in Eq. (7) remain unchanged. For example, the quasi-local term $\tilde{\mathcal{V}}_x(t)$ transforms to

$$
\mathcal{V}_x(t) = \frac{U}{L^2} \sum_{\{k_i\}} e^{ix(k_1+k_2-k_3-k_4)} \Big( f_R(\{k_i\}, t) L_{k_1}^\dagger R_{k_2}^\dagger R_{k_3} R_{k_4} + f_L(\{k_i\}, t) L_{k_1}^\dagger L_{k_2}^\dagger L_{k_3} R_{k_4} \Big), \tag{28}
$$

where in the rotated frame $f_R(\{k_i\}, t) = \Big( \alpha_{k_1}(t)\alpha_{k_2}^*(t) + \beta_{k_1}(t)\beta_{k_2}^*(t) \Big) \alpha_{k_3}(t)\beta_{k_4}(t) e^{i\theta_{k_1}(t)}$ and $f_L(\{k_i\}, t) = \alpha_{k_1}(t)\beta_{k_2}(t) e^{i(\theta_{k_1}(t)+\theta_{k_2}(t)-\theta_{k_3}(t))} \Big( \alpha_{k_3}^*(t)\alpha_{k_4}(t) + \beta_{k_3}^*(t)\beta_{k_4}(t) \Big)$. The states, however, obtain fast oscillating phases,

$$
|\Psi_i(t)\rangle = e^{i(N_{L,i}-N_{R,i})\Delta t/2}|\tilde{\Psi}_i(t)\rangle \quad ; \quad \big|\Psi_f(t)\big\rangle = e^{i(N_{L,f}-N_{R,f})\Delta t/2}|\tilde{\Psi}_f(t)\rangle, \tag{29}
$$

because the initial (final) state contains $N_{R,i}$ ($N_{R,f}$) creation operators $R_k^\dagger$ (particles in the $R$-band) and $N_{L,i}$ ($N_{L,f}$) operators $L_k^\dagger$ (particles in the $L$-band). Since $N_{L,f} = N_{L,i} + 1$ and $N_{R,f} = N_{R,i} - 1$ the matrix element appearing in Fermi's golden rule (22) obtains a phase,

$$
\langle \tilde{\Psi}_f(t)|\tilde{\mathcal{V}}_x(t)|\tilde{\Psi}_i(t)\rangle = \langle \tilde{\Psi}_f(t)|\mathcal{V}_x(t)|\Psi_i(t)\rangle e^{i\Delta t}. \tag{30}
$$

For the contribution of the nearby terms in Eq. (22) this means

$$
\Gamma_{near} \le \frac{2r^*+1}{L\mathcal{T}} \sum_f \sum_{x=1}^{L} \left| \int_0^{\mathcal{T}} dt \left\langle \Psi_f(t) \right| \mathcal{V}_x(t) \left| \Psi_i(t) \right\rangle e^{i\Delta t} \right|^2. \tag{31}
$$

From now on we will only work in the basis of the transformed frame where the diagonal elements of $\mathcal{H}_0$ are of order $\mathcal{O}(W)$, and the system is time-dependent with a large effective effective frequency $\Delta \gg W$. In this way we cast the low-frequency driving problem to a high-frequencey driving one, akin to the situation studied in [46]. However, in the present case an additional driving frequency $\omega$ is present in addition to the large frequency $\Delta$.

We will now proceed to derive a bound on the excitation rate from Eq. (31) for $\Delta \gg \omega, W, U$. To this end we use the identity $e^{i\Delta t} = (-i\partial_t)^M \Delta^{-M} e^{i\Delta t}$ to convert the fast oscillations $e^{i\Delta t}$ inside the integral in Eq. (31) into a small prefactor controlled by a large denominator $1/\Delta^M$ for a large power M (the value of $M$ will be determined in Eq. (50)). Using this identity and then performing integration by parts to move the derivative to the matrix element, we obtain

$$
\int_0^{\mathcal{T}} dt \left\langle \Psi_f(t) \right| \mathcal{V}_x(t) \left| \Psi_i(t) \right\rangle e^{i\Delta t} = \int_0^{\mathcal{T}} dt \left\langle \Psi_f(t) \right| \mathcal{V}_x(t) \left| \Psi_i(t) \right\rangle \left( \frac{-i\partial_t}{\Delta} \right)^M e^{i\Delta t}
$$
$$
= G_i(\mathcal{T}, M, x) + G_b(\mathcal{T}, M, x), \tag{32}
$$

where

$$
G_i(\mathcal{T}, M, x) \equiv \int_0^{\mathcal{T}} dt \, e^{i\Delta t} \left( \frac{i\partial_t}{\Delta} \right)^M \left\langle \Psi_f(t) \right| \mathcal{V}_x(t) \left| \Psi_i(t) \right\rangle, \tag{33}
$$

and the boundary terms are given by

$$
G_b(\mathcal{T}, M, x) \equiv \left[ \sum_{m=0}^{M-1} \frac{-i}{\Delta} e^{i\Delta t} \left( \frac{i\partial_t}{\Delta} \right)^m \left\langle \Psi_f(t) \right| \mathcal{V}_x(t) \left| \Psi_i(t) \right\rangle \right]_{t=0}^{\mathcal{T}}. \tag{34}
$$

To simplify notation we define the operator $\mathcal{O}_x^M(t)$ via

$$
\left\langle \Psi_f(t) \right| \mathcal{O}_x^M(t) \left| \Psi_i(t) \right\rangle \equiv \left( \frac{i\partial_t}{\Delta} \right)^M \left\langle \Psi_f(t) \right| \mathcal{V}_x(t) \left| \Psi_i(t) \right\rangle. \tag{35}
$$

Note that this condition on a single matrix element does not uniquely define the operator $\mathcal{O}_x^M(t)$. However, in the following it is sufficient to consider one such operator that satisfies Eq. (35) for Floquet eigenstates $|\Psi_{i/f}(t)\rangle$. In section 4.2.3 we specify our choice for the operator $\mathcal{O}_x^M(t)$.

### 4.2.2 Excitation rate

We apply expression (32) to Eq. (31) and take this as starting point to calculate the excitation probability. This implies taking a square of (32). Using a triangle inequality we bound the cross-terms by the sum of the squares,

$$
\Gamma_{near} \leq \frac{2(2r^*+1)}{L\mathcal{T}} \sum_f \sum_{x=1}^{L} \left( |G_i(\mathcal{T}, M, x)|^2 + |G_b(\mathcal{T}, M, x)|^2 \right). \tag{36}
$$

The boundary terms in Eq. (34), for each $x$, are oscillatory functions of the total time $\mathcal{T}$. In appendix B we show that they are bounded by a constant of order $\mathcal{O}(U/\Delta)^2$. Due to the $1/\mathcal{T}$ prefactor in the scattering rate the contribution of the boundary terms $|\mathcal{G}_b(\mathcal{T}, M, x)|^2$ in Eq. (36) decay to zero at long times. The persistent heating comes from the remaining integral in Eq. (33). In the same appendix we show that in leading order these integrals give a $\mathcal{T}$-independent contribution in Eq. (36) and depend on the norm of the operator $\mathcal{O}_x^M$ defined in Eq. (35). Altogether we obtain as bound on the excitation rate

$$
\Gamma_{near} \leq (2r^*+1) \frac{8\pi}{\omega} \|\mathcal{O}_x^M\|^2 + \frac{8(2r^*+1)}{\mathcal{T}} \left( \frac{U}{\Delta} \right)^2, \tag{37}
$$

with the usual operator norm, $\|\mathcal{O}_x^M\|^2 \equiv sup_{|\Psi|=1} \langle\Psi|\,\mathcal{O}_x^{M\dagger}\mathcal{O}_x^M\,|\Psi\rangle$. The second term gives a small offset at short times because, by assumption, we have $U \ll \Delta$. At long times $\mathcal{T}$ this term decays to zero. The rate of persistent heating is thus controlled by the norm of the operator $\mathcal{O}_x^M$. In the next section we will show that this norm is exponentially small:

$$\|\mathcal{O}_x^M(t)\| \le 5KU e^{-\mu\Delta/\mathcal{E}}, \tag{38}$$

where $K$ is a dimensionless constant and $\mathcal{E}$ is the largest of the small energy scales, $\mathcal{E} = \max\{K_U U, K_E W, K_\omega \omega\}$. The constants $K_{U,E,\omega}$ are of order unity and depend on properties of the Floquet single-particle eigenstates.

### 4.2.3  Bound on the operator norm

All that remains is to derive a bound on the norm of the operator $\mathcal{O}_x^M$ in Eq. (35). We first consider the effect of taking one derivative of the matrix element, then we iterate the procedure. The many-body states $\left|\Psi_{i/f}(t)\right\rangle$ are defined as solutions of the Schrödinger equation with the Hamiltonian $\mathcal{H}_0(t) + \mathcal{V}_{intra}(t)$, therefore the matrix elements $\left\langle\Psi_f(t)\right|\mathcal{V}_x(t)|\Psi_i(t)\rangle$ obey

$$\Delta^{-1}i\partial_t \left\langle\Psi_f(t)\right|\mathcal{V}_x(t)|\Psi_i(t)\rangle = \left\langle\Psi_f(t)\right|[\Delta^{-1}\mathcal{V}_x(t), \mathcal{H}_0(t) + \mathcal{V}_{intra}(t)] + \left(\Delta^{-1}i\partial_t\mathcal{V}_x(t)\right)|\Psi_i(t)\rangle, \tag{39}$$

with the interband scattering operator $\mathcal{V}_x(t)$ in the transformed frame from Eq. (28). There are three different contributions in Eq. (39): a commutator of $\mathcal{V}_x(t)$ with the quadratic Hamiltonian $\mathcal{H}_0(t)$, a commutator with the band-preserving interaction $\mathcal{V}_{intra}(t)$, and a time-derivative of the interband interaction operator $\mathcal{V}_x(t)$. We consider these terms individually to show that each yields one of the energy scales $\omega, W, U$, together with the prefactor $\Delta^{-1}$ this gives an operator with a small norm. Then we iterate the process for $M$ derivatives to obtain a factor $(U^q W^p \omega^{M-p-q})/\Delta)^M$, where $p$ and $q$ are positive integers. The calculation for $\mathcal{V}_x^{\mathrm{RR}\to\mathrm{RL}}$ and $\mathcal{V}_x^{\mathrm{RL}\to\mathrm{LL}}$ is very similar in each of these three contributions, therefore we will demonstrate only the calculation for $\mathcal{V}_x^{\mathrm{RR}\to\mathrm{RL}}$.

**Commutator with quadratic Hamiltonian:**  To calculate the commutators in Eq. (39) the relevant parts of $\mathcal{V}_x$ in Eq. (28) are the operators $L_{k_1}^\dagger R_{k_2}^\dagger R_{k_3} R_{k_4}$ and $L_{k_1}^\dagger L_{k_2}^\dagger L_{k_3} R_{k_4}$; the remainder are scalar functions. Here we focus on the $L_{k_1}^\dagger R_{k_2}^\dagger R_{k_3} R_{k_4}$ term. The second term is computed in the same way.

We first examine the commutator of $\mathcal{V}_x$ with $\mathcal{H}_0$, the quadratic single-particle Hamiltonian (24). The off-diagonal elements in Eq. (24), which contain $R_k^\dagger L_k$ or $L_k^\dagger R_k$, change the occupation numbers of the two bands. When commuted with $\mathcal{V}_x$, the $R_k^\dagger L_k$-term gives an operator that preserves the occupation numbers, and the $L_k^\dagger R_k$-term gives an operator that changes the occupation numbers by two. However, since the occupation numbers of $|\Psi_i\rangle$ and $\left|\Psi_f\right\rangle$ differ by exactly one, $N_{L,f} - N_{L,i} = 1$, the resulting matrix element vanishes:

$$\left\langle\Psi_f\right|\left[L_{k_1}^\dagger R_{k_2}^\dagger R_{k_3} R_{k_4}, \sum_k e_{12,k} R_k^\dagger L_k + e_{21,k} L_k^\dagger R_k\right]|\Psi_i\rangle = 0. \tag{40}$$

Therefore we only need to consider commutators involving the diagonal elements of $\mathcal{H}_0$ in (24). The commutator gives

$$\left[\Delta^{-1}\mathcal{V}_x^{\mathrm{RR}\to\mathrm{RL}}, W\sum_k e_{11,k} R_k^\dagger R_k + e_{22,k} L_k^\dagger L_k\right] = \tag{41}$$
$$\frac{U}{\Delta}\frac{W}{L^2}\sum_{\{k_i\}} e^{ix(k_1+k_2-k_3-k_4)} f_{\mathrm{R}}(\{k_i\}, t)(-e_{22,k_1} - e_{11,k_2} + e_{11,k_3} + e_{11,k_4}) L_{k_1}^\dagger R_{k_2}^\dagger R_{k_3} R_{k_4}.$$

This is a sum of four terms which all, following from the periodicity of $f_R(\{k_i\}, t)$ and $e_{11,k}(t)$ $[e_{22,k}(t)]$, have a prefactor that is periodic in all $k_i$ and oscillates with frequency $\omega$. In the rotating frame we showed that $|E_{11,k}|, |E_{22,k}| \leq \frac{W}{2}$, i.e. $|e_{11,k}|, |e_{22,k}| \leq \frac{1}{2}$. Hence the magnitude of each term that is generated by the commutator in Eq. (41) is bounded by $UW/2\Delta$, a reduction by a factor of $W/2\Delta$ as compared to the original magnitude of $\mathcal{V}_x^{\mathrm{RR}\to\mathrm{RL}}$. The same argument applies to the commutator of $\mathcal{V}_x^{\mathrm{RL}\to\mathrm{LL}}$ with $\mathcal{H}_0$.

**Commutator with band-conserving interaction:** The commutator of the quasi-local operator $\mathcal{V}_x^{\mathrm{RR}\to\mathrm{RL}}(t)$ with the intraband interaction operator from Eq. (12) contains commutations with $\mathcal{V}_{intra}^{RR}(t)$, $\mathcal{V}_{intra}^{LR}(t)$, and $\mathcal{V}_{intra}^{LL}(t)$. As an example, the commutator of $V_x^{\mathrm{RR}\to\mathrm{RL}}(t)$ with $\mathcal{V}_{intra}^{RR}(t)$ in Eq. (13) is given by

$$[\Delta^{-1}\mathcal{V}_x^{\mathrm{RR}\to\mathrm{RL}}(t), \mathcal{V}_{intra}^{RR}(t)] = \frac{U}{\Delta}\frac{U}{L^4}\sum_{\{k_i,q_i\}}e^{ix(k_1+k_2-k_3-k_4)}\sum_{x'=1}^{L}e^{ix'(q_1+q_2-q_3-q_4)} \tag{42}$$
$$f_R(\{k_i\},t)g_{RR}(\{q_i\},t)[L_{k_1}^\dagger R_{k_2}^\dagger R_{k_3}R_{k_4}, R_{q_1}^\dagger R_{q_2}^\dagger R_{q_3}R_{q_4}].$$

After some algebra (see appendix C.1 for details), the commutator in Eq. (42) is reduced to a sum of six terms, each involving a string of six creation/annihilation operators (i.e., corresponding to three-particle interaction). One example term is

$$\frac{U}{\Delta}\frac{U}{L^3}\sum_{\{k_i\}}e^{ix(k_1+k_2+k_3-k_4-k_5-k_6)}h(\{k_i\},t)L_{k_1}^\dagger R_{k_2}^\dagger R_{k_3}^\dagger R_{k_4}R_{k_5}R_{k_6}, \tag{43}$$

where

$$h(\{k_i\},t) = \left(\alpha_{k_1}\alpha_{k_2}^* + \beta_{k_1}\beta_{k_2}^*\right)\alpha_{k_3}^*\beta_{k_4}\alpha_{k_5}\beta_{k_6}\alpha_{-k_3+k_5+k_6}\beta_{-k_3+k_5+k_6}^* e^{i\theta_{k_1}}. \tag{44}$$

This resulting term $h(\{k_i\}, t)$ is periodic in all $k_i$, as is $f_R(\{k_i\}, t)$. Together with the exponential factor $e^{ix(k_1+k_2+k_3-k_4-k_5-k_6)}$ this ensures the resulting operator is a sum of quasi-local operators. The commutators with $\mathcal{V}_{intra}^{LR}(t)$ and $\mathcal{V}_{intra}^{LL}(t)$ yield similar terms as in Eq. (43). Compared with the magnitude of the four-point interaction terms in $\mathcal{V}_x$, the magnitudes of the six-point interaction terms generated by the commutator in Eq. (42) are suppressed by a factor $U/\Delta$. In appendix C.1 we show that overall there are 24 such terms. Additionally there are 24 more similar terms coming from the commutator of $\mathcal{V}_x^{\mathrm{RL}\to\mathrm{LL}}$ with $\mathcal{V}_{intra}$.

**Time-derivative of the operator:** The third term in Eq. (39) contains the explicit time-derivative of the operator $\mathcal{V}_x(t)$ in Eq. (28). The time-dependent parts therein are the Floquet single-particle creation and annihilation operators in the rotated frame, $R_k^\dagger(t), L_k^\dagger(t), R_k(t), L_k(t)$, and the functions $\alpha_k(t), \beta_k(t)$. In appendix C.2 we show that the derivative of $\mathcal{V}_x^{\mathrm{RR}\to\mathrm{RL}}(t)$ is a sum of four terms of the form

$$\frac{\omega}{\Delta}\frac{U}{L^2}\sum_{\{k_i\}}e^{i(k_1+k_2-k_3-k_4)x}G(\{k_i\},t)L_{k_1}^\dagger(t)R_{k_2}^\dagger(t)R_{k_3}(t)R_{k_4}(t), \tag{45}$$

where in all terms $G(\{k_i\}, t)$ is periodic in all $k_i$. The function $G(\{k_i\}, t)$ is a product of functions $\alpha_k(t), \beta_k(t)$ and their derivatives, hence it is periodic in $t$ with frequency $\omega$. We further show in appendix C.2 that its Fourier transformation with respect to time has the largest coefficients at small frequencies much less than $\Delta$, and the Fourier coefficients decay exponentially beyond that regime. We will return to this fact later. Note that Eq. (45) contains the same single-particle Floquet operators as $\mathcal{V}_x^{\mathrm{RR}\to\mathrm{RL}}(t)$. A similar calculation can be done to obtain the time-derivative of $\mathcal{V}_x^{\mathrm{RL}\to\mathrm{LL}}(t)$.

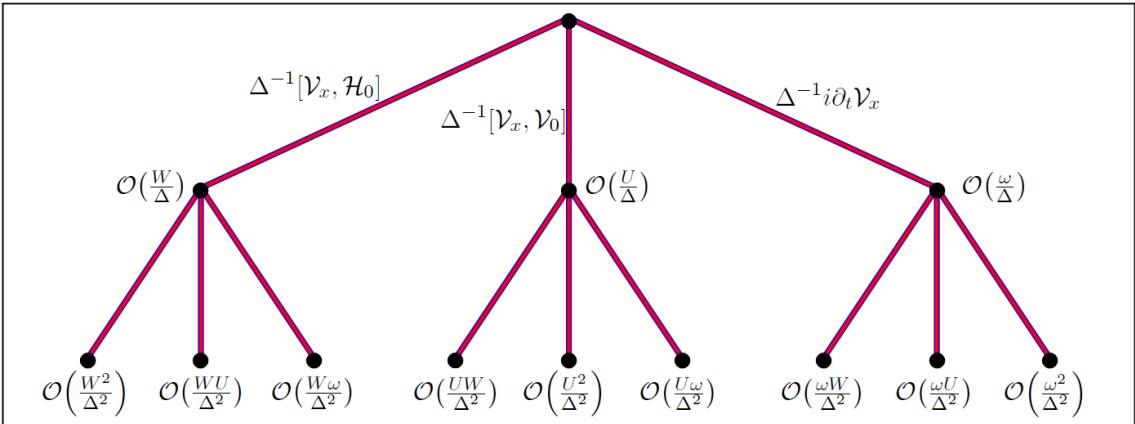

Figure 4: The process of taking iterated derivatives of the matrix element can be visualized in a tree structure. We start with 1 term $\mathcal{V}_x(t)$ on the left. From there we follow the three different branches where each creates multiple terms with an additional ratio of energy scales $\frac{W}{\Delta}$, $\frac{U}{\Delta}$, or $\frac{\omega}{\Delta}$. From each of these points there emerge three new branches corresponding to the three different terms in the derivative which yield a combination of the energy scales. After $M$ steps there are $3^M$ branches with multiple operators each and combined energy scales $\frac{U^q W^p \omega^{M-p-q}}{\Delta^M}$.

**Iterated derivatives of the matrix element:**   So far we discussed the effects of taking one derivative $i\partial_t$ of the matrix element in Eq. (39). To calculate the norm of the operator $\mathcal{O}_x^M(t)$ defined in Eq. (35) we need to iterate this step $M$ times. We showed that all three contributions to the derivative yield operators which are similar to $\mathcal{V}_x(t)$ in Eq. (28), in the sense that they contain a string of creation and annihilation operators $R_k^\dagger, L_k^\dagger, R_k, L_k$, they overall move one particle from the $R$-band to the $L$-band, and their prefactors are periodic functions in all $k_i$.

When we iterate the derivative, in each step we start with a sum of terms like $\mathcal{V}_x(t)$ in Eq. (28). From each term we obtain several new terms akin to equations (41), (43) and (45). Starting from one term with $n$ single-particle Floquet state creation/annihilation operators $R_k^\dagger, L_k^\dagger, R_k, L_k$, the three different contributions from Eq. (39) give:

- Commutator with $\mathcal{H}_0$: $n$ terms with $n$ creation/annihilation operators, each multiplied by an additional factor $W/\Delta$,

- Commutator with $\mathcal{V}_{intra}$: $6n$ terms with $n+2$ creation/annihilation operators, each multiplied by an additional factor $U/\Delta$,

- Derivative $\Delta^{-1} i\partial_t$: $n$ terms with $n$ creation/annihilation operators, and each multiplied by an additional factor $\omega/\Delta$

Therefore the total number of terms created by one term with $n$ single-particle operators is $n+6n+n=8n$. This process is visualized in Fig. 4. In the $m$-th step we obtain terms involving strings with different numbers of single-particle operators: the commutator with the intraband interaction $\mathcal{V}_{intra}(t)$ increases that number $n$ by 2, while the commutator with $\mathcal{H}_0(t)$ and the time-derivative leave $n$ unchanged.

As we can see from the discussion above, the longer the operator string (i.e., the larger the number $n$), the larger the number of additional terms that are generated in each iteration. Hence, to obtain an upper bound on the total number $\mathcal{N}$ of terms in $\mathcal{O}_x^M(t)$ we can replace $n$ in the $m$-th iteration by the maximal possible length of an operator string generated after $m-1$ iterations of the derivative, $n_{max}(m) = 2m + 2$. The iteration begins with two quartic

terms $\mathcal{V}_x^{\text{RR}\to\text{RL}}$ and $\mathcal{V}_x^{\text{RL}\to\text{LL}}$, each involving $n = 4$ creation/annihilation operators. Hence the total number $\mathcal{N}$ of terms after $M$ steps is bounded by

$$\mathcal{N} \le 2 \prod_{m=1}^{M} 8 n_{max}(m) = 2 \prod_{m=1}^{M} (16m + 16) = 2(M + 1)! 16^M \le 5(16M)^M. \tag{46}$$

In the last step we use the bound $(M + 1)! \le \frac{5}{2}M^M$, which is valid for all $M > 0$.

**Operator norm after iterated derivatives:**  Now we are in the position to write down the explicit form of the operator $\mathcal{O}_x^M(t)$ in Eq. (35) and calculate a bound for the norm $\|\mathcal{O}_x^M(t)\|$ in Eq. (38). After $M$ iterations of taking derivatives $\mathcal{O}_x^M(t)$ is a sum of $\mathcal{N}$ terms. Each contains a prefactor $U^q W^p \omega^{M-p-q}/\Delta^M$. The integers $p, q$ depend on the path that was taken in Fig. 4, i.e. $p, q$ count the number of times the commutator is taken with $\mathcal{H}_0$ and $\mathcal{V}_{intra}$, respectively. Note that each path leads to multiple terms. We use the label $\eta$ to index all of the terms arising from all of the different paths. Each term contains an additional dimensionless prefactor $\mathcal{F}_\eta(\{k_i\}, t)$ which is periodic in all $k_i$. Hence we write the operator $\|\mathcal{O}_x^M(t)\|$ in the form

$$\begin{aligned} \mathcal{O}_x^M(t) &= \sum_{\eta=1}^{\mathcal{N}} U \frac{U^q W^p \omega^{M-p-q}}{\Delta^M} \frac{1}{L^{n/2}} \\ &\quad \sum_{\{k_i\}} e^{ix(k_1+\ldots+k_{n/2}-k_{n/2+1}-\ldots-k_n)} \mathcal{F}_\eta(\{k_i\}, t) L_{k_1}^\dagger \ldots R_{k_n}, \end{aligned} \tag{47}$$

where in the above sum, $p$ and $q$ are determined by the path corresponding to each term $\eta$. We discuss the properties of $\mathcal{F}_\eta(\{k_i\}, t)$ in detail in appendix C.3, and show that the norm of this operator is bounded by

$$\|\mathcal{O}_x^M(t)\| \le 5KU \left(\frac{16M\mathcal{E}}{\Delta}\right)^M. \tag{48}$$

The parameter $\mathcal{E}$ is the largest of the renormalized energy scales

$$\mathcal{E} = \max\{K_U U, K_E W, K_\omega \omega\}. \tag{49}$$

Here $K, K_U, K_E, K_\omega$ are dimensionless constants. They depend on the properties of the Floquet single-particle states. In general $K, K_U, K_E$ are of order $\mathcal{O}(1)$ for a Hamiltonian which is a sum of local terms. For $\omega > W$ the constant $K_\omega$ also is of order 1, $K_\omega = \mathcal{O}(1)$, while for $W > \omega$ it goes as $K_\omega = \mathcal{O}(\frac{W}{\omega})$. In either case the resulting energy scale $\mathcal{E}$ in Eq. (49) is proportional to one of the small energy scales $U, W, \omega$. In appendix C.3 we derive explicit expressions for these constants.

So far $M$ was a free parameter; Eq. (32) holds for all $M$. Hence we can freely choose this parameter. By differentiating we see that the tightest bound is reached for $M = \frac{\mu\Delta}{\mathcal{E}}$ with $\mu = (16e)^{-1}$. With this optimal choice of $M$, Eq. (48) becomes

$$\|\mathcal{O}_x^M(t)\| \le 5KU e^{-\mu\Delta/\mathcal{E}}. \tag{50}$$

Our main result from this section is the exponential bound on the operator norm in Eq. (50). The main message here is that this norm is exponentially small if $\Delta \gg W, U, \omega$. To obtain the final result for the excitation rate from the nearby terms we apply the result (50) to Eq. (37):

$$\Gamma_{near} \le 200\pi(2r^* + 1)K^2 \frac{U^2}{\omega} e^{-2\mu\Delta/\mathcal{E}} + \frac{8(2r^* + 1)}{\mathcal{T}} \left(\frac{U}{\Delta}\right)^2. \tag{51}$$

Together with the exponentially suppressed contribution from the distant terms, Eq. (19), this gives an upper bound on the total rate of change of the densities $n_{R,L}$ of particles in the $R$- and $L$-band $\Gamma = \Gamma_{near} + \Gamma_{dist}$. For times $T \ll \mathcal{T} \ll r^*/v$ the first term in Eq. (51) is the dominant contribution, and the effect from the distant terms in Eq. (19) is negligible. In thermodynamically large systems $r^*$ can be chosen sufficiently large to have a long window during which $T \ll \mathcal{T} \ll r^*/v$. In this time interval the rate of creating excitations is exponentially small. Hence, during this long time window a prethermal quasi-steady state persists.

## 5  Summary and discussion

In this paper we showed that in slowly driven systems, assuming the hierarchy of energy scales $\Delta \gg U, W, \omega$, the rate at which particles transition from one band to the other is exponentially small in the large ratio $\Delta/\mathcal{E}$ where $\mathcal{E} = \max(U, W, \omega)$. The intraband thermalization rate is not suppressed by $\Delta/\mathcal{E}$. Therefore we expect thermalization within each band to occur rapidly compared to the timescale associated with the decay of the imbalance of the band populations. This leads to a prethermal quasi-steady state in which all single-particle Floquet eigenstates within one band are populated equally while there is a population imbalance between the two bands. The inverse of the excitation rate in Eq. (51) gives a lower bound on the lifetime of the quasi-steady prethermal state. Although we were motivated by dynamics of topological pumps, we stress that our results hold for slowly-driven systems under very general assumptions. We only require that all terms in the Hamiltonian and the interaction are local operators, and the band gap $\Delta$ is the largest energy scale in comparison to the drive frequency $\omega$, the band width $W$, and the interaction strength $U$. The dimensionless factor which multiplies the ratio of the energy scales in the exponent only depends on the properties of the single-particle Floquet eigenstates; in appendix C.3 we provide an explicit calculation of this factor. Generally, a small localization length of the Floquet-Wannier functions leads to a large factor in the exponent and strong suppression of the excitation rate. This localization length also depends on the ratios $\Delta/W$ and $\Delta/\omega$. Therefore there are subleading corrections in the final result in Eq. (51) which enhance the suppression of the transition rate as $\Delta/\mathcal{E}$ increases.

We obtained an upper bound for the rate $\Gamma$ of particles being transferred from the $R$- to the $L$-band. If the $L$-band in the initial state $|\Psi_i\rangle$ is not empty, the opposite process of particles transitioning from the $L$- to the $R$-band is also possible. Hence the decay rate of the population imbalance between the $R$- and the $L$-bands will be smaller than the bound we calculated. If we assume an initial state where the $L$-band is empty there is no backflow of particles. In this case we can derive a tighter bound on the excitation rate, see appendix D. This tighter bound is similar to Eq. (51) with the replacement of $\mu \to \tilde{\mu} = 2\mu$. We expect that the rate of exciting particles is largest in the absence of a backflow of particles, i.e. when the $L$-band is empty. This is in agreement with numerical simulations in [57]. Hence we may expect that the tighter bound also holds when the $L$-band is partially filled.

In appendix E we provide generalizations of our bound. First we generalize to any number of spatial dimensions, and second we generalize the interaction. In Eq. (2) we assume a 2-particle interaction which only acts on particles on the same site. In appendix E we take a general short-range 2-particle interaction and derive a similar exponential bound. In the same spirit it is also possible to assume short-range interactions between multiple particles. Third we generalize to systems with an arbitrary number of bands. In multi-band systems we only require that the bands fall into groups which are separated by a large bandgap $\Delta$. All in all we can apply our results to a wide class of slowly-driven systems. Another interesting question is whether a similar bound holds for a system of bosons, or whether bunching effects lead to qualitatively different behavior.

Following the original argument by Thouless [53], half filling is required in order to obtain a gapped state and thereby to obtain a quantized pump. This was observed in cold atom experiments with bosons [54] and fermions [55], as well as in quantum dots [62]. Here we show that in the presence of inter-particle interactions, there exist exponentially long-lived chiral prethermal states at arbitrary filling fractions (see also [57]). Thus our results opens new possibilities for observing topological Floquet bandstructures in quantum many body systems.

## Acknowledgements

N.L., T.G. and E.B. acknowledge support from the European Research Council (ERC) under the European Union Horizon 2020 Research and Innovation Programme (Grant Agreement No. 639172). T.G. was in part supported by an Aly Kaufman Fellowship at the Technion. T.G. acknowledges funding from the Institute of Science and Technology (IST) Austria, and from the European Union's Horizon 2020 research and innovation programme under the Marie Skłodowska-Curie Grant Agreement No. 754411. N.L. acknowledges support from the People Programme (Marie Curie Actions) of the European Unions Seventh Framework 546 Programme (FP7/20072013), under REA Grant Agreement No. 631696, and by the Israeli Center of Research Excellence (I-CORE) Circle of Light funded by the Israel Science Foundation (Grant No. 1802/12). M.R. gratefully acknowledges the support of the European Research Council (ERC) under the European Union Horizon 2020 Research and Innovation Programme (Grant Agreement No. 678862). M.R. acknowledges the support of the Villum Foundation. M.R. and E.B. acknowledge support from CRC 183 of the Deutsche Forschungsgemeinschaft.

## A   Lattice model for Thouless' pump

To obtain the plots of the band structure we use the same model for a 1-dimensional topological charge pump as in Ref. [57]. The tight-binding Hamiltonian is

$$\tilde{\mathcal{H}}_0(t) = -J(t)\sum_j c_{j,A}^\dagger c_{j,B} - J'(t)\sum_j c_{j,B}^\dagger c_{j+1,A} + h.c. + \mu(t)\sum_j (c_{j,A}^\dagger c_{j,A} - c_{j,B}^\dagger c_{j,B}). \quad (52)$$

This is a tight-binding Hamiltonian with alternating nearest-neighbor hopping $J(t)$ and $J'(t)$, and alternating on-site potential $\pm\mu(t)$. For the figures we set $J(t) = J_0 + \delta J(t)$ and $J'(t) = J_0 - \delta J(t)$ with $J_0 = 1$ and $\delta J(t) = 1.2\sin(\omega t)$. The on-site potential is chosen as $\mu(t) = 2.5\cos(\omega t)$. In Fig. 2 the frequency is $\omega = 0.15$.

## B   Constant excitation rate and offset

Here we show that from Eq. (36) we obtain an excitation rate which is constant in time plus a decreasing offset.

The full form of Eq. (36) is

$$
\begin{aligned}
\Gamma_{near}(\mathcal{T}) \leq\ & \frac{2(2r^*+1)}{L\mathcal{T}}\sum_f\sum_{x=1}^L \left|\left[\sum_{m=0}^{M-1}\frac{-i}{\Delta}e^{i\Delta t}\left(\frac{i\partial_t}{\Delta}\right)^m \langle\Psi_f(t)|\mathcal{V}_x(t)|\Psi_i(t)\rangle\right]_{t=0}^{\mathcal{T}}\right|^2 \\
& + \frac{2(2r^*+1)}{L\mathcal{T}}\sum_f\sum_{x=1}^L \left|\int_0^{\mathcal{T}} dt\, e^{i\Delta t}\left(\frac{i\partial_t}{\Delta}\right)^M \langle\Psi_f(t)|\mathcal{V}_x(t)|\Psi_i(t)\rangle\right|^2. \quad (53)
\end{aligned}
$$

The boundary terms in the first line give a function which is oscillating in $\mathcal{T}$ but not increasing. The prefactor $1/\mathcal{T}$ ensures that this term vanishes at long times. Every derivative $i\partial_t$ of the matrix element yields an energy $\omega, U, W$. Together with the small factor $1/\Delta$ we see that these terms give a small offset in which the main contribution is the $m = 0$ term. Full justification of this qualitative argument follows from the derivation in section 4.2.3. Hence for the boundary terms we estimate

$$
\begin{aligned}
\Gamma_b(\mathcal{T}) &\leq \frac{2(2r^*+1)}{L\mathcal{T}} \sum_{x=1}^{L} \sum_f \quad \left| \left[ \frac{-i}{\Delta} e^{i\Delta t} \left\langle \Psi_f(t) \right| \mathcal{V}_x(t) \left| \Psi_i(t) \right\rangle \right]_{t=0}^{\mathcal{T}} \right|^2 \\
&\leq \frac{4(2r^*+1)}{L\mathcal{T}} \sum_{x=1}^{L} \frac{1}{\Delta^2} \sum_f \quad \Big( \langle \Psi_i(0) | \mathcal{V}_x^\dagger(0) | \Psi_f(0) \rangle \langle \Psi_f(0) | \mathcal{V}_x(0) | \Psi_i(0) \rangle \\
&\qquad\qquad\qquad\qquad + \langle \Psi_i(\mathcal{T}) | \mathcal{V}_x^\dagger(\mathcal{T}) | \Psi_f(\mathcal{T}) \rangle \langle \Psi_f(\mathcal{T}) | \mathcal{V}_x(\mathcal{T}) | \Psi_i(\mathcal{T}) \rangle \Big) \\
&\leq \frac{8(2r^*+1)}{\mathcal{T}} \left( \frac{U}{\Delta} \right)^2 .
\end{aligned}
\tag{54}
$$

In the first step we apply a triangle inequality, so that we only obtain terms with the same time. In the second step we use $\sum_f \left| \Psi_f(t) \right\rangle \left\langle \Psi_f(t) \right| = \mathbb{1}$ within the subspace of fixed particle numbers $N_{R,f}$ and $N_{L,f}$, and the norm is $\| \mathcal{V}_x^\dagger(t) \mathcal{V}_x(t) \| = U^2$ as in section 4.1. Therefore the offset to the excitation probability goes, in leading order, as $(U/\Delta)^2$ which is small for a large bandgap and weak interactions.

Hence, the dominant contribution to heating and eventual thermalization is the part which comes from the second line in Eq. (53). Here we want to show that this part is independent of time $\mathcal{T}$, i.e. the system has a constant heating rate. To simplify notation we define the operator $\mathcal{O}_x^M(t)$ implicitly by (cf. Eq. (35) in the main text)

$$
\left\langle \Psi_f(t) \right| \mathcal{O}_x^M(t) \left| \Psi_i(t) \right\rangle \equiv \left( \frac{i\partial_t}{\Delta} \right)^M \left\langle \Psi_f(t) \right| \mathcal{V}_x(t) \left| \Psi_i(t) \right\rangle .
\tag{55}
$$

This definition is not unique, however in the following we only need to consider one operator $\mathcal{O}_x^M(t)$ which fulfills Eq. (55). From the periodicity of $\mathcal{V}_x(t)$ follows that the operator $\mathcal{O}_x^M(t)$ is periodic in time, and we decompose it into its Fourier modes $\mathcal{O}_x^{M\,m}$. Similarly $\left| \Psi_{i/f}(t) \right\rangle$ are Floquet eigenstates of the Hamiltonian $\mathcal{H}_0 + \mathcal{V}_{intra}$ with quasienergies $\epsilon_{i/f}$, so we can write them in their Fourier modes,

$$
|\Psi_i(t)\rangle = e^{-i\epsilon_i t} \sum_l e^{il\omega t} |\Psi_i^l\rangle \quad , \quad \left| \Psi_f(t) \right\rangle = e^{-i\epsilon_f t} \sum_n e^{in\omega t} |\Psi_f^n\rangle.
\tag{56}
$$

Rewriting the second line in Eq. (53) in terms of these Fourier modes gives

$$
\begin{aligned}
\Gamma_i(\mathcal{T}) &\leq \frac{2(2r^*+1)}{L\mathcal{T}} \sum_f \sum_{x=1}^{L} \left| \int_0^{\mathcal{T}} dt \sum_{l,m,n} e^{-i(\epsilon_f - \epsilon_i - \Delta + (l-m-n)\omega)t} \langle \Psi_f^l | \mathcal{O}_x^{M\,m} | \Psi_i^n \rangle \right|^2 \\
&= \frac{2(2r^*+1)}{L\mathcal{T}} \sum_f \sum_{x=1}^{L} \left| \mathcal{T} \sum_{l,m,n} \mathrm{sinc}\left( E_{l,m,n} \frac{\mathcal{T}}{2} \right) \langle \Psi_f^l | \mathcal{O}_x^{M\,m} | \Psi_i^n \rangle \right|^2 .
\end{aligned}
\tag{57}
$$

In the last line we set $E_{l,m,n} = \epsilon_f - \epsilon_i - \Delta + (l-m-n)\omega$, and $\mathrm{sinc}(x) = \frac{\sin(x)}{x}$ is the cardinal sine function. It is sharply peaked at $x = 0$ and has a decaying envelope $x^{-1}$. Following the standard derivation of Fermi's golden rule for a continuum of final states we argue that the cardinal sine is negligible outside the interval $x \in [-\pi, \pi]$, which are the two central zeroes. Hence we only need to consider $E_{l,m,n} \in [-\frac{2\pi}{\mathcal{T}}, \frac{2\pi}{\mathcal{T}}]$. For $\mathcal{T} > 2T$ not for all quasi-energies $\epsilon_f \in [0, \omega]$ exist $l, m, n$ so that $E_{l,m,n}$ falls into this interval. In the thermodynamic limit there is a continuum of final states. By assumption they are thermalized, which means they are

indistinguishable and obtain a constant density of states, $\rho(\epsilon_f) = \rho$ for $\epsilon_f \in [0, \omega]$. [27] Hence we can convert the sum over final states into an integral, $\sum_f \to \int d\epsilon_f \rho(\epsilon_f)$, with a constant density of states $\rho(\epsilon_f) = \rho$ for $\epsilon_f \in [0, \omega]$. Thus the fraction of final states that we need to consider is $\frac{2T}{\mathcal{T}}$ of all states, and we continue:

$$
\begin{aligned}
\Gamma_i(\mathcal{T}) &\leq \frac{2(2r^*+1)}{L\mathcal{T}} \sum_{x=1}^{L} \int d\epsilon_f \rho(\epsilon_f) \left| \mathcal{T} \sum_{l,m,n} \text{sinc}\left( E_{l,m,n} \frac{\mathcal{T}}{2} \right) \langle \Psi_f^l | \mathcal{O}_x^{Mm} | \Psi_i^n \rangle \right|^2 \\
&\approx \frac{2(2r^*+1)}{L\mathcal{T}} \mathcal{T}^2 L \frac{4\pi\rho}{\mathcal{T}} \left| \sum_{l,m,n} \langle \Psi_f^l | \mathcal{O}_{x_0}^m | \Psi_i^n \rangle \right|^2 \\
&= (2r^*+1) 8\pi\rho \left| \langle \Psi_f(0) | \mathcal{O}_{x_0}(0) | \Psi_i(0) \rangle \right|^2.
\end{aligned}
\tag{58}
$$

Here we used translational invariance, i.e. the sum over all sites becomes $\sum_{x=1}^{L} \to L$ and we consider the operator at one example point $x_0$. Because the states are indistinguishable the quasi-local operator $\mathcal{O}_x^M(t)$ acts on all of them in a similar way, and the matrix element does not depend on the particular choice of $\Psi_f$. To obtain the matrix element and the density of states we calculate in two ways:

$$
\begin{aligned}
\sum_f \left| \langle \Psi_f(0) | \mathcal{O}_{x_0}(0) | \Psi_i(0) \rangle \right|^2 &= \int d\epsilon_f \rho(\epsilon_f) \left| \langle \Psi_f(0) | \mathcal{O}_{x_0}(0) | \Psi_i(0) \rangle \right|^2 \\
&= \rho\omega \left| \langle \Psi_f(0) | \mathcal{O}_{x_0}(0) | \Psi_i(0) \rangle \right|^2, \\
\sum_f \left| \langle \Psi_f(0) | \mathcal{O}_{x_0}(0) | \Psi_i(0) \rangle \right|^2 &= \langle \Psi_i(0) | \mathcal{O}_{x_0}^\dagger(0) \mathcal{O}_{x_0}(0) | \Psi_i(0) \rangle \leq \|\mathcal{O}_{x_0}\|^2.
\end{aligned}
\tag{59}
$$

Combined these show that the matrix element times the density of states is bounded by the norm of the operator. Applying this to Eq. (58) we arrive at

$$
\Gamma_i \leq (2r^*+1) \frac{8\pi}{\omega} \|\mathcal{O}_{x_0}(0)\|^2.
\tag{60}
$$

## C  Details for the calculations in section 4.2.3

In this appendix we summarize the technical details for the calculations in section 4.2.3. These are required to derive the operator $\mathcal{O}_x^M(t)$ in Eq. (47), and subsequently derive a bound on its norm.

### C.1  Commutator of the interaction with the intraband operator

In this section we show the explicit calculation of the commutator of $\mathcal{V}_x(t)$ with the intraband operator $\mathcal{V}_{intra}(t)$ in section 4.2.3. We start with Eq. (42), the commutator of $\mathcal{V}_x^{\text{RR}\to\text{RL}}$ with $\mathcal{V}_{intra}^{\text{RR}}$. It contains two terms of eight Floquet single-particle operators each, in different order. By using the canonical fermionic anticommutation relations of the Floquet single-particle operators,

$$
\{R_k^\dagger, R_q\} = \delta_{k,q} = \{L_k^\dagger, L_q\} \quad ; \quad \{R_k^\dagger, R_q^\dagger\} = \{R_k, R_q\} = 0 = \{L_k^\dagger, R_q\} = \{L_k^\dagger, R_q^\dagger\},
\tag{61}
$$

we bring the operators in the same order to cancel the terms with eight operators. This requires six non-trivial commutations according to (61). In the end we arrive at

$$[L_{k_1}^\dagger R_{k_2}^\dagger R_{k_3} R_{k_4}, R_{q_1}^\dagger R_{q_2}^\dagger R_{q_3} R_{q_4}] = \tag{62}$$

$$L_{k_1}^\dagger R_{k_2}^\dagger R_{k_3} R_{q_2}^\dagger R_{q_3} R_{q_4} \delta_{k_4,q_1} \quad - \quad L_{k_1}^\dagger R_{k_2}^\dagger R_{k_4} R_{q_2}^\dagger R_{q_3} R_{q_4} \delta_{k_3,q_1} +$$

$$L_{k_1}^\dagger R_{k_2}^\dagger R_{q_1}^\dagger R_{k_3} R_{q_3} R_{q_4} \delta_{k_4,q_2} \quad - \quad L_{k_1}^\dagger R_{k_2}^\dagger R_{q_1}^\dagger R_{k_4} R_{q_3} R_{q_4} \delta_{k_3,q_2} +$$

$$L_{k_1}^\dagger R_{q_1}^\dagger R_{q_2}^\dagger R_{q_4} R_{k_3} R_{k_4} \delta_{k_2,q_3} \quad - \quad L_{k_1}^\dagger R_{q_1}^\dagger R_{q_2}^\dagger R_{q_3} R_{k_3} R_{k_4} \delta_{k_2,q_4}.$$

The result are six terms with six operators each. This we apply to Eq. (42) and calculate for one example term:

$$\frac{U}{\Delta}\frac{U}{L^4} \sum_{x'=1}^L \sum_{\{k_i,q_i\}} -f_R(\{k_i\},t) g_{RR}(\{q_i\},t) L_{k_1}^\dagger R_{k_2}^\dagger R_{q_1}^\dagger R_{k_4} R_{q_3} R_{q_4} \delta_{k_3,q_2} = \tag{63}$$

$$\frac{U}{\Delta}\frac{U}{L^3} \sum_{\{k_i\}} -e^{ix(k_1+k_2-k_3-k_4)} \left(\alpha_{k_1}\alpha_{k_2}^* + \beta_{k_1}\beta_{k_2}^*\right) \alpha_{k_3}\beta_{k_4} e^{i\theta_{k_1}}$$

$$\sum_{\{q_i\}} L^{-1} \sum_{x'=1}^L e^{ix'(q_1+q_2-q_3-q_4)} \alpha_{q_1}^* \beta_{q_2}^* \alpha_{q_3}\beta_{q_4} L_{k_1}^\dagger R_{k_2}^\dagger R_{q_1}^\dagger R_{k_4} R_{q_3} R_{q_4} \delta_{k_3,q_2} =$$

$$\frac{U}{\Delta}\frac{U}{L^3} \sum_{\{k_i\}} -e^{ix(k_1+k_2-k_3-k_4)} \left(\alpha_{k_1}\alpha_{k_2}^* + \beta_{k_1}\beta_{k_2}^*\right) \alpha_{k_3}\beta_{k_4} e^{i\theta_{k_1}}$$

$$\sum_{\{q_i\}} \delta_{k_3,q_2} \delta_{q_2,-q_1+q_3+q_4} \alpha_{q_1}^* \beta_{q_2}^* \alpha_{q_3}\beta_{q_4} L_{k_1}^\dagger R_{k_2}^\dagger R_{q_1}^\dagger R_{k_4} R_{q_3} R_{q_4} =$$

$$\frac{U}{\Delta}\frac{U}{L^3} \sum_{k_1,k_2,k_4,q_1,q_3,q_4} -e^{ix(k_1+k_2+q_1-k_4-q_3-q_4)} \left(\alpha_{k_1}\alpha_{k_2}^* + \beta_{k_1}\beta_{k_2}^*\right) \alpha_{-q_1+q_3+q_4}\beta_{k_4} e^{i\theta_{k_1}}$$

$$\alpha_{q_1}^* \beta_{-q_1+q_3+q_4}^* \alpha_{q_3}\beta_{q_4} L_{k_1}^\dagger R_{k_2}^\dagger R_{q_1}^\dagger R_{k_4} R_{q_3} R_{q_4}.$$

In the first step we use the definitions of the functions $f_R(\{k_i\},t)$ and $g_{RR}(\{q_i\},t)$ from equations (28) and (12). In the second step we solve the sum over $x'$ which yields a $\delta$-function over the momenta $q_i$. In the third step we solve the sums over $q_1$ and $k_2$ by using the $\delta$-functions. All six terms from Eq. (62) have a similar shape.

Next we turn to the commutator with $\mathcal{V}_{intra}^{LR}$. Commuting the single-particle Floquet operators gives

$$[L_{k_1}^\dagger R_{k_2}^\dagger R_{k_3} R_{k_4}, L_{q_1}^\dagger R_{q_2}^\dagger L_{q_3} R_{q_4}] = \tag{64}$$

$$L_{q_1}^\dagger L_{k_1}^\dagger R_{k_2}^\dagger L_{q_3} R_{k_4} R_{q_4} \delta_{k_3,q_2} \quad - \quad L_{q_1}^\dagger L_{k_1}^\dagger R_{k_2}^\dagger L_{k_3} R_{q_3} R_{q_4} \delta_{k_4,q_2} +$$

$$L_{q_1}^\dagger R_{q_2}^\dagger L_{q_3} L_{k_1}^\dagger R_{k_3} R_{k_4} \delta_{k_2,q_4} \quad - \quad L_{q_1}^\dagger R_{q_2}^\dagger R_{k_2}^\dagger R_{k_3} R_{k_4} R_{q_4} \delta_{k_1,q_3}.$$

These are four additional terms, however the prefactor $g_{LR}$ in Eq. (14) is a sum of 4 terms. Therefore this gives a total of 16 new terms which are all of similar structure as Eq. (63).

The last contribution is the commutator with $\mathcal{V}_{intra}^{LL}$, for which we calculate

$$[L_{k_1}^\dagger R_{k_2}^\dagger R_{k_3} R_{k_4}, L_{q_1}^\dagger L_{q_2}^\dagger L_{q_3} L_{q_4}] = \tag{65}$$

$$L_{q_1}^\dagger L_{q_2}^\dagger L_{q_4} R_{k_2}^\dagger R_{k_3} R_{k_4} \delta_{k_1,q_3} \quad - \quad L_{q_1}^\dagger L_{q_2}^\dagger L_{q_3} R_{k_2}^\dagger R_{k_3} R_{k_4} \delta_{k_1,q_4}.$$

These are 2 more terms with the same structure as Eq. (63). Hence in total the commutator of $\mathcal{V}_x^{RR \to RL}$ with the intraband interaction $\mathcal{V}_{intra}$ generates 24 new terms. The same calculation with $\mathcal{V}_x^{RL \to LL}$ yields the same result, the commutator generates 24 more terms.

## C.2 Explicit time-derivative

This subsection deals with the explicit time-derivative of the operator $\mathcal{V}_x(t)$. To take the derivative of $\mathcal{V}_x^{RR \to RL}(t)$ we need to calculate

$$\frac{i\partial_t}{\Delta} \left(\left(\alpha_{k_1}(t)\alpha_{k_2}^*(t) + \beta_{k_1}(t)\beta_{k_2}^*(t)\right) \alpha_{k_3}(t)\beta_{k_4}(t) e^{i\theta_{k_1}(t)} L_{k_1}^\dagger(t) R_{k_2}^\dagger(t) R_{k_3}(t) R_{k_4}(t)\right). \tag{66}$$

We use the product rule of the derivative to separate this into four terms, where in each we take the derivative of terms with one $k_i$ only. We have e.g. for $k_4$

$$\frac{i\partial_t}{\Delta}\left(\beta_{k_4}(t)R_{k_4}(t)\right) = \frac{i\partial_t}{\Delta}\left(\beta_{k_4}(t)\alpha_{k_4}^*(t)A_{k_4} + \beta_{k_4}(t)\beta_{k_4}^*(t)B_{k_4}\right), \tag{67}$$

where we used the relation between Floquet eigenstates and sublattice basis, Eq. (26). These are products of two functions which by themselves are not periodic in time, but their products are:

$$\begin{aligned}\beta_k(t+T)\alpha_k^*(t+T) &= e^{-i(\epsilon_k+\Delta/2)T}\beta_k(t)e^{i(\epsilon_k+\Delta/2)T}\alpha_k^*(t) \\ &= \beta_k(t)\alpha_k^*(t) \equiv \sum_l\left(\beta_k\alpha_k^*\right)^{(l)}e^{-il\omega t}.\end{aligned} \tag{68}$$

In the last step we expand this product into its temporal Fourier series. With its help we calculate the derivative in Eq. (67):

$$\frac{i\partial_t}{\Delta}\left(\beta_{k_4}(t)R_{k_4}(t)\right) = \frac{\omega}{\Delta}\sum_l le^{-il\omega t}\left(\left(\beta_{k_4}\alpha_{k_4}^*\right)^{(l)}A_{k_4} + \left(\beta_{k_4}\beta_{k_4}^*\right)^{(l)}B_{k_4}\right). \tag{69}$$

We rewrite this in terms of the Floquet eigenstate operators in the rotating frame to arrive at

$$\begin{aligned}\frac{i\partial_t}{\Delta}\left(\beta_{k_4}R_{k_4}\right) &= \frac{\omega}{\Delta}\sum_l le^{-il\omega t}\left(\left(\beta_{k_4}\alpha_{k_4}^*\right)^{(l)}\alpha_{k_4}(t) + \left(\beta_{k_4}\beta_{k_4}^*\right)^{(l)}\beta_{k_4}(t)\right)R_{k_4} \\ &\quad + \frac{\omega}{\Delta}\sum_l le^{-il\omega t}\left(-\left(\beta_{k_4}\alpha_{k_4}^*\right)^{(l)}\beta_{k_4}^*(t) + \left(\beta_{k_4}\beta_{k_4}^*\right)^{(l)}\alpha_{k_4}^*(t)\right)e^{-i\theta_{k_4}(t)}L_{k_4}.\end{aligned} \tag{70}$$

The $L_{k_4}$ term in Eq. (70), combined with the operators $L_{k_1}^\dagger R_{k_2}^\dagger R_{k_3}$ in Eq. (66), yields an operator which preserves the band occupation numbers. Since the states $|\Psi_i\rangle$ and $|\Psi_f\rangle$ have different occupation numbers this matrix element is zero, similar to the argument in Eq. (40). Hence the term involving $L_{k_4}$ in Eq. (70) can be neglected. Thus, from the explicit action of the time derivative we obtain a new term which contains the same single-particle operators as $\mathcal{V}_x^{\text{RR}\to\text{RL}}(t)$, with an additional prefactor $\frac{\omega}{\Delta}$. We perform analogous calculations for the terms with $k_1, k_2, k_3$ in Eq. (66), where each gives a new term involving the same single-particle operators. Overall we obtain four new terms which are all of the form

$$\frac{\omega U}{\Delta L^2}\sum_{\{k_i\}}e^{i(k_1+k_2-k_3-k_4)x}G(\{k_i\},t)L_{k_1}^\dagger(t)R_{k_2}^\dagger(t)R_{k_3}(t)R_{k_4}(t), \tag{71}$$

where for all cases $G(\{k_i\},t)$ is periodic in all $k_i$. In the example from Eq. (70),

$$\begin{aligned}G(\{k_i\},t) &= \left(\alpha_{k_1}\alpha_{k_2}^* + \beta_{k_1}\beta_{k_2}^*\right)\alpha_{k_3}\beta_{k_4}e^{i\theta_{k_1}} \\ &\quad \sum_l le^{-il\omega t}\left(\left(\beta_{k_4}\alpha_{k_4}^*\right)^{(l)}\alpha_{k_4} + \left(\beta_{k_4}\beta_{k_4}^*\right)^{(l)}\beta_{k_4}\right).\end{aligned} \tag{72}$$

The functions $\tilde{\alpha}_k(t), \tilde{\beta}_k(t)$ describing the single particle Floquet states in the original (non-rotating) frame, defined in Eq. (9), have most of their spectral weight in the Fourier modes with frequency $l\omega \approx \Delta/2$. In the rotating frame, the spectral weights of $\alpha_k(t), \beta_k(t)$ are shifted to the slow modes around $l \approx 0$ with decay for large $l$ as $e^{-A|l|^{3/2}}$ [57]. Therefore, the sum over the absolute value of all modes $l$ is finite. We will return to this sum in section C.3.

## C.3 Bound on the operator norm

In this appendix we derive the final bound on the operator $\mathcal{O}_x^M(t)$ in section 4.2.3. We start with the expression in Eq. (47),

$$
\mathcal{O}_x^M(t) = \sum_{\eta=1}^{\mathcal{N}} U \frac{U^q W^p \omega^{M-p-q}}{\Delta^M} \frac{1}{L^{n/2}}
$$
$$
\sum_{\{k_i\}} e^{ix(k_1+\ldots+k_{n/2}-k_{n/2+1}-\ldots-k_n)} \mathcal{F}_\eta(\{k_i\},t) L_{k_1}^\dagger R_{k_2}^\dagger \ldots R_{k_n}. \tag{73}
$$

$\mathcal{F}_\eta(\{k_i\},t)$ is a periodic function for all $k_i$ because it is a product of periodic functions $\alpha_k$, $\beta_k$, $e_{11,k}$, and $e_{22,k}$. That implies the Fourier coefficients in the expansion

$$
\mathcal{F}_\eta(\{k_i\},t) = \sum_{\{z_i\}} \overline{\mathcal{F}}_\eta(\{z_i\},t) e^{i(z_1 k_1 + \ldots - z_n k_n)} \tag{74}
$$

go to zero exponentially fast for large $|z_i|$. The same applies to the Fourier series of the other periodic functions,

$$
\alpha_k = \sum_y \overline{\alpha}_y e^{iyk} \quad ; \quad \beta_k = \sum_y \overline{\beta}_y e^{iyk} \quad ; \quad e_{11,k} = \sum_y \overline{e}_y e^{iyk}, \tag{75}
$$

the coefficients go to zero at large $|y|$. We apply these expansion to Eq. (73) and rewrite the Floquet state operators in the site basis $\{A_{x_i}, B_{x_i}\}$ according to definition (9):

$$
\mathcal{O}_x^M(t) = \sum_{\eta=1}^{\mathcal{N}} U \frac{U^q W^p \omega^{M-p-q}}{\Delta^M} \sum_{\{x_i,y_i,z_i\}} \frac{1}{L^n} \sum_{\{k_i\}} e^{ik_1(x+x_1-y_1+z_1)+\ldots+ik_n(-x+x_n-y_n-z_n)}
$$
$$
\overline{\mathcal{F}}_\eta(\{z_i\},t)(-\overline{\beta}_{y_1}^* A_{x_1}^\dagger + \overline{\alpha}_{y_1}^* B_{x_1}^\dagger) \ldots (\overline{\alpha}_{y_n}^* A_{x_n} + \overline{\beta}_{y_n}^* B_{x_n}) \tag{76}
$$
$$
= \sum_{\eta=1}^{\mathcal{N}} U \frac{U^q W^p \omega^{M-p-q}}{\Delta^M} \sum_{\{y_i,z_i\}} \overline{\mathcal{F}}_\eta(\{z_i\},t)
$$
$$
(-\overline{\beta}_{y_1}^* A_{y_1-x-z_1}^\dagger + \overline{\alpha}_{y_1}^* B_{y_1-x-z_1}^\dagger) \ldots (\overline{\alpha}_{y_n}^* A_{x+y_n+z_n} + \overline{\beta}_{y_n}^* B_{x+y_n+z_n}).
$$

The momenta $k_i$ only appear in the exponent thus in the second step we can sum over all of them and obtain Kronecker-$\delta$s. We use these to solve the sums over all sites $x_i$. Note that in this expression any operator far away from $x$ has an exponentially small prefactor because either $\overline{\mathcal{F}}_\eta$ or the $\overline{\alpha}, \overline{\beta}$ are small in that case. This is a manifestation of locality of the operator $\mathcal{O}_x^M(t)$. The norm for each on-site operator is $\|A_{x_i}\| = \|B_{x_i}\| = 1$, so we can estimate the norm of $\mathcal{O}_x^M(t)$ by taking absolute values of each individual factor. As a result the sums over $\{y_i,z_i\}$ decouple:

$$
\|\mathcal{O}_x^M(t)\| \leq \sum_{\eta=1}^{\mathcal{N}} U \frac{U^q W^p \omega^{M-p-q}}{\Delta^M} \sum_{\{y_i,z_i\}} (|\overline{\beta}_{y_1}| + |\overline{\alpha}_{y_1}|) \ldots (|\overline{\alpha}_{y_n}| + |\overline{\beta}_{y_n}|)|\overline{\mathcal{F}}_\eta(\{z_i\},t)|. \tag{77}
$$

As we argued above, the absolute values of the coefficients $\overline{\alpha}_y, \overline{\beta}_y, \overline{e}_y$ in the Fourier series (75) go to zero exponentially fast for large $|y|$. Hence the sums over their absolute value are finite and we define the dimensionless constants

$$
K_\alpha = \max_t \sum_y |\overline{\alpha}_y(t)| \quad ; \quad K_\beta = \max_t \sum_y |\overline{\beta}_y(t)| \quad ; \quad K_E = \max_t \sum_y |\overline{e}_y(t)|. \tag{78}
$$

Similarly we know that the coefficients $\overline{\mathcal{F}}_\eta(\{z_i\}, t)$ decay exponentially at large $|z_i|$ and therefore the sum $\sum_{\{z_i\}} |\overline{\mathcal{F}}_\eta(\{z_i\}, t)|$ converges. We calculate it iteratively. The main idea hereby is that each step in the iterated derivative causes the same change to this sum. In the initial step $M = 0$ we have $\mathcal{N} = 2$ operators with $\mathcal{F}_1(\{k_i\}, t) = f_R(\{k_i\}, t) = (\alpha_{k_1} \alpha^*_{k_2} + \beta_{k_1} \beta^*_{k_2}) \alpha_{k_3} \beta_{k_4} e^{i\theta_{k_1}}$ and $\mathcal{F}_2(\{k_i\}, t) = f_L(\{k_i\}, t)$. For now we focus on the terms from $f_R(\{k_i\}, t)$. The summation over $\{z_i\}$ is bounded by

$$\sum_{\{z_i\}} |\overline{f}_R(\{z_i\}, t)| = \sum_{\{z_i\}} |(\overline{\alpha}_{z_1} \overline{\alpha}^*_{z_2} + \overline{\beta}_{z_1} \overline{\beta}^*_{z_2}) \overline{\alpha}_{z_3} \overline{\beta}_{z_4}| \leq K_\alpha K_\beta (K_\alpha^2 + K_\beta^2). \tag{79}$$

In the main text in section 4.2.3 we discussed the three different kinds of terms that appear from taking the derivative of the matrix element, and there are multiple terms of each kind. The index $\eta$ labels all these terms including which branch we follow during the iterated derivatives. In the following we calculate the change to the sum over the coefficients $\overline{\mathcal{F}}_\eta(\{z_i\})$ in Eq. (77) after taking one derivative.

**Commutator with the quadratic Hamiltonian:**  After one commutation with the quadratic Hamiltonian we obtain four terms where in each the function $\mathcal{F}_\eta(\{k_i\}, t)$ is of the form

$$\mathcal{F}_\eta(\{k_i\}, t) = e_{11,k_4}(\alpha_{k_1} \alpha^*_{k_2} + \beta_{k_1} \beta^*_{k_2}) \alpha_{k_3} \beta_{k_4} = \sum_{\{z_i\}} \overline{\mathcal{F}}_\eta(\{z_i\}, t) e^{i(z_1 k_1 + z_2 k_2 - z_3 k_3 - z_4 k_4)}. \tag{80}$$

To express the Fourier coefficients $\overline{\mathcal{F}}_\eta(\{z_i\})$ we use the Fourier series in (75). From this we derive

$$\overline{\mathcal{F}}_\eta(\{z_i\}, t) = (\overline{\alpha}_{z_1} \overline{\alpha}^*_{z_2} + \overline{\beta}_{z_1} \overline{\beta}^*_{z_2}) \overline{\alpha}_{z_3} \sum_z \overline{\beta}_{z_4 - z} \overline{e}_z. \tag{81}$$

The momentum $k_4$ appears twice in the expression for $\mathcal{F}_\eta(\{k_i\}, t)$ which manifests itself in a convolution of the indices $z_4$ and $z$. However this has no effect on the summation over the absolute values:

$$\begin{aligned}
\sum_{\{z_i\}} |\overline{\mathcal{F}}_\eta(\{z_i\}, t)| &= \sum_{\{z_i\}} |(\overline{\alpha}_{z_1} \overline{\alpha}^*_{z_2} + \overline{\beta}_{z_1} \overline{\beta}^*_{z_2}) \overline{\alpha}_{z_3} \sum_z \overline{\beta}_{z_4 - z} \overline{e}_z| \\
&\leq K_\alpha (K_\alpha^2 + K_\beta^2) \sum_{z, z_4} |\overline{\beta}_{z_4 - z} \overline{e}_z| \\
&\leq K_\alpha K_\beta K_E (K_\alpha^2 + K_\beta^2).
\end{aligned} \tag{82}$$

Compared with the bound in Eq. (79) this only differs by an additional factor $K_E = \sum_z |\overline{e}_z|$. So in every iteration of taking the commutator with the quadratic Hamiltonian the bound on the sum over coefficients $|\overline{\mathcal{F}}_\eta(\{z_i\}, t)|$ is multiplied by a factor $K_E$.

**Commutator with the intraband operator:**  In a very similar fashion the commutator with the intraband term creates a convolution in the Fourier series but with more terms involved. The calculation follows the same lines. Taking the example term in Eq. (43) we have

$$\mathcal{F}_\eta(\{k_i\}, t) = h(\{k_i\}, t) = \left(\alpha_{k_1} \alpha^*_{k_2} + \beta_{k_1} \beta^*_{k_2}\right) \alpha_{-k_3 + k_5 + k_6} \beta_{k_4} \alpha^*_{k_3} \beta^*_{-k_3 + k_5 + k_6} \alpha_{k_5} \beta_{k_6}. \tag{83}$$

In the Fourier series this gives a convolution of three momenta $k_2, k_3, k_4$ and the five terms which contain them. The coefficients are

$$\overline{\mathcal{F}}_\eta(\{z_i\}, t) = (\overline{\alpha}_{z_1} \overline{\alpha}^*_{z_2} + \overline{\beta}_{z_1} \overline{\beta}^*_{z_2}) \overline{\beta}_{z_4} \sum_{w_1, w_2} \overline{\alpha}_{w_1} \overline{\alpha}^*_{-z_3 - w_1 - w_2} \overline{\beta}^*_{w_2} \overline{\alpha}_{z_5 - w_1 - w_2} \overline{\beta}_{z_6 - w_1 - w_2}, \tag{84}$$

and summation over all terms gives

$$
\begin{aligned}
\sum_{\{z_i\}} |\overline{\mathcal{F}}_\eta(\{z_i\}, t)| &\le K_\beta(K_\alpha^2 + K_\beta^2) \sum_{w_1, w_2, z_3, z_5, z_6} |\overline{\alpha}_{w_1} \overline{\alpha}^*_{-z_3-w_1-w_2} \overline{\beta}^*_{w_2} \overline{\alpha}_{z_5-w_1-w_2} \overline{\beta}_{z_6-w_1-w_2}| \\
&\le K_\alpha^3 K_\beta^3 (K_\alpha^2 + K_\beta^2).
\end{aligned}
\tag{85}
$$

Compared with the bound in Eq. (79) there is an additional factor $K_\alpha^2 K_\beta^2$. However, commutation with the intraband operator also increases the number of Floquet single-particle operators by 2. As we have seen in Eq. (77) these sums decouple and the contribution to the norm from each pair of Floquet operators $R_k^\dagger R_{k'}$ is $(K_\alpha + K_\beta)^2$. Hence in every iteration of commuting with the intraband operator the bound on the sum over coefficients is multiplied by $K_\alpha^2 K_\beta^2 (K_\alpha + K_\beta)^2$.

**Time derivative of the operator:** The function $\mathcal{F}_\eta$ for one example term when taking the time derivative is calculated in Eq. (70):

$$
\mathcal{F}_\eta(\{k_i\}, t) = \left(\alpha_{k_1} \alpha^*_{k_2} + \beta_{k_1} \beta^*_{k_2}\right) \alpha_{k_3} \sum_l l e^{-il\omega t} \left(\left(\beta_{k_4} \alpha^*_{k_4}\right)^{(l)} \alpha_{k_4} + \left(\beta_{k_4} \beta^*_{k_4}\right)^{(l)} \beta_{k_4}\right).
\tag{86}
$$

The Fourier series in $k_1, k_2, k_3$ is simple, in $k_4$ this gives a convolution of three terms. We obtain the Fourier coefficients as

$$
\begin{aligned}
\overline{\mathcal{F}}_\eta(\{z_i\}, t) =\ & \left(\overline{\alpha}_{z_1} \overline{\alpha}^*_{z_2} + \overline{\beta}_{z_1} \overline{\beta}^*_{z_2}\right) \overline{\alpha}_{z_3} \sum_l l e^{-il\omega t} \\
& \sum_{w_1, w_2} \left(\left(\overline{\beta}_{w_1} \overline{\alpha}^*_{w_2}\right)^{(l)} \overline{\alpha}_{z_4-w_1-w_2} + \left(\overline{\beta}_{w_1} \overline{\beta}^*_{w_2}\right)^{(l)} \overline{\beta}_{z_4-w_1-w_2}\right).
\end{aligned}
\tag{87}
$$

The summation over all $\{z_i\}$ gives

$$
\begin{aligned}
\sum_{\{z_i\}} |\overline{\mathcal{F}}_\eta(\{z_i\}, t)| \le\ & K_\beta(K_\alpha^2 + K_\beta^2) \sum_{z_4, w_1, w_2} \\
& \left|\sum_l l \left(\left(\overline{\beta}_{w_1} \overline{\alpha}^*_{w_2}\right)^{(l)} \overline{\alpha}_{z_4-w_1-w_2} + \left(\overline{\beta}_{w_1} \overline{\beta}^*_{w_2}\right)^{(l)} \overline{\beta}_{z_4-w_1-w_2}\right)\right|.
\end{aligned}
\tag{88}
$$

As discussed in the main text, the product $\alpha_k(t)\beta_k^*(t)$ is periodic in $t$ (Eq. (68)). Therefore the absolute value of the coefficients in the Fourier series over time $t$, $(\alpha_k \beta_k^*)^{(l)}$, goes to zero exponentially fast at large $l$. Starting from a Zener tunneling problem, [57] shows that these coefficients decay as $e^{-A|l|^{3/2}}$. Therefore the sum in Eq. (88) is finite and with the definition of a constant

$$
\kappa \equiv K_\alpha^{-1}(K_\alpha^2 + K_\beta^2)^{-1} \max_t \sum_{y_1, y_2, y_3} \left|\sum_l l \left(\overline{\beta}_{y_1} \overline{\alpha}^*_{y_2}\right)^{(l)} \overline{\alpha}_{y_3-y_1-y_2} + \left(\overline{\beta}_{y_1} \overline{\beta}^*_{y_2}\right)^{(l)} \overline{\beta}_{y_3-y_1-y_2}\right|
\tag{89}
$$

we get

$$
\sum_{\{z_i\}} |\overline{\mathcal{F}}_\eta(\{z_i\}, t)| \le K_\alpha K_\beta \kappa (K_\alpha^2 + K_\beta^2)^2.
\tag{90}
$$

This differs from the result in Eq. (79) by a factor $\kappa(K_\alpha^2 + K_\beta^2)$. Every iteration of the application of the time derivative produces this additional factor.

To summarize this: after $M$ iterations of taking the derivative of the matrix element there are

- the initial factor $K \equiv K_\alpha K_\beta (K_\alpha^2 + K_\beta^2)(K_\alpha + K_\beta)^4$ from Eq. (79) and the initial 4 Floquet operators as an overall prefactor.

- a factor $K_E^p$ from $p$ commutators with the quadratic Hamiltonian.

- a factor $K_U^q \equiv K_\alpha^{2q} K_\beta^{2q} (K_\alpha + K_\beta)^{2q}$ from $q$ commutators with the intraband operator.

- a factor $K_\omega^{M-p-q} \equiv \kappa^{M-p-q} (K_\alpha^2 + K_\beta^2)^{M-p-q}$ from $M - p - q$ applications of the time-derivative to the operator.

Now we are ready to apply this to the norm in Eq. (77):

$$\|\mathcal{O}_x^M(t)\| \leq \sum_{\eta=1}^{\mathcal{N}} KU \frac{(K_U U)^q (K_E W)^p (K_\omega \omega)^{M-p-q}}{\Delta^M} \leq 5 KU \left( \frac{16 M \mathcal{E}}{\Delta} \right)^M. \tag{91}$$

In the last step we defined the energy scale $\mathcal{E}$ as the maximum of all the involved renormalized energy scales,

$$\mathcal{E} = \max\{K_U U, K_E W, K_\omega \omega\}. \tag{92}$$

The expression (91) holds for all $M$ and we can apply it to Eq. (47) in the main text.

# D Initial state with empty L-band

In section 5 we mention that we obtain a tighter bound if in the initial state $|\Psi_i\rangle$ the $L$-band is empty. This is the case, up to corrections of order $\mathcal{O}(\omega/\Delta)^2$ [69], at short times $\mathcal{T}$ if the system is initialized in the ground state of the static Hamiltonian before switching on the drive. There are two changes to the derivation, which mostly affect the counting of terms in Eq. (46). First we can neglect the intraband interaction that requires an $L$-particle in the initial state, $\mathcal{V}_x^{\text{RL}\to\text{LL}}$. This operator can't act on the state $|\Psi_i\rangle$. Secondly, when calculating the commutator of $\mathcal{V}_x$ with the intraband term $\mathcal{V}_{intra}$ in section 4.2.3 and appendix C.1 the counting of terms changes. The commutator $[\mathcal{V}_x^{\text{RR}\to\text{RL}}, \mathcal{V}_{intra}^{\text{RR}}]$ is unchanged, from there we obtain six new terms. Next we commute with $\mathcal{V}_{intra}^{\text{LR}}$ defined in Eq. (14). The matrix element, reduced to the operators, is

$$\langle \Psi_f | [L_{k_1}^\dagger R_{k_2}^\dagger R_{k_3} R_{k_4}, L_{q_1}^\dagger R_{q_2}^\dagger L_{q_3} R_{q_4}] | \Psi_i \rangle. \tag{93}$$

The part of the commutator where the term to the right first acts on the initial state is zero because by assumption the $L$-band is empty and $L_{q_3}$ removes a particle from that band. In the second part $L_{k_1}^\dagger$ creates the only particle in the $L$-band, hence $L_{q_3}$ has to act on this particle which gives $\delta_{k_1, q_3}$. With this we arrive at the same shape as in Eq. (63), however the prefactor function $g_{\text{LR}}(\{q_i\}, t)$ only contains four pieces, hence this gives four additional terms similar to (63). The commutator with $\mathcal{V}_{intra}^{\text{LL}}$ does not yield any contribution because $\mathcal{V}_x |\Psi_i\rangle$ has only one particle in the $L$-band and $\mathcal{V}_{intra}^{\text{LL}}$ can only act on states with at least 2 $L$-particles. Thus we overall obtain ten new terms. When iterating these steps the commutator with $\mathcal{V}_{intra}^{\text{LR}}$ always only gives four terms, independent of the number $n$ of single-particle Floquet operators. The commutator with $\mathcal{V}_{intra}^{\text{RR}}$ gives $2n - 2$ terms, which in total makes $2n + 2$ terms from the commutator.

This changes the counting of operators in section 4.2.3 and the bound on the total number in Eq. (46). We now obtain at most $n + (2n + 2) + n = 4n + 2$ terms at each step of taking the iterated derivatives. The total number of terms after $M$ steps therefore is bounded by

$$\tilde{\mathcal{N}} \leq \prod_{m=1}^{M} 4 n_{max}(m) + 2 = \prod_{m=1}^{M} (8m + 10) \leq \frac{5}{2} (8M)^M. \tag{94}$$

This is in comparison to the bound $\mathcal{N} \leq 2(16M)^M$ in the main text. This changes the final result from Eq. (51) to

$$\Gamma_{near} \leq 50\pi(2r^* + 1)K^2\frac{U^2}{\omega}e^{-2\tilde{\mu}\Delta/\mathcal{E}} + 8(2r^* + 1)L\left(\frac{U}{\Delta}\right)^2, \tag{95}$$

with $\tilde{\mu} = (8e)^{-1} = 2\mu$. Hence the overall prefactor is divided by 4, and the factor $\tilde{\mu}$ in the exponent is doubled which leads to stronger suppression.

## E   Generalization to arbitrary dimensions and multi-band systems

In this appendix we generalize the calculation of the excitation rate in section 4.2 to multi-band systems in any number of dimensions, and to interactions that do not only act within a single unit cell. We consider a non-interacting Hamiltonian with $N$ bands in the basis of Floquet single-particle eigenstates $\tilde{\psi}_\nu(\mathbf{k})$,

$$\tilde{\mathcal{H}}_0(t) = \sum_{\mathbf{k}} \tilde{\psi}_\nu^\dagger(\mathbf{k}, t)\tilde{E}_{\nu,\tau}(\mathbf{k}, t)\tilde{\psi}_\tau(\mathbf{k}, t). \tag{96}$$

Here and in the following we use Einstein summation for greek indices with $\nu, \tau = 1, ..., N$ running over all bands, and we use bold font $\mathbf{k}$ for vectors in $d$ dimensions. The Floquet states are related to the sublattice basis by a unitary transformation akin to Eq. (9),

$$\tilde{\psi}^\dagger(\mathbf{k}, t) = \tilde{\alpha}(\mathbf{k}, t)A^\dagger(\mathbf{k}), \tag{97}$$

where $\tilde{\alpha}(\mathbf{k}, t)$ is a unitary $N \times N$ matrix. The eigenstates and the matrix elements in the Hamiltonian follow the same periodicity properties as in Eq. (9) in the main text,

$$\begin{aligned}
\tilde{\alpha}_{\nu,\tau}(\mathbf{k} + \mathbf{G}, t) &= \tilde{\alpha}_{\nu,\tau}(\mathbf{k}, t) &= e^{i\epsilon_{\nu,\mathbf{k}}T}\tilde{\alpha}_{\nu,\tau}(\mathbf{k}, t + T), \\
\tilde{E}(\mathbf{k} + \mathbf{G}, t) &= \tilde{E}(\mathbf{k}, t) &= \tilde{E}(\mathbf{k}, t + T),
\end{aligned} \tag{98}$$

for any inverse lattice vector $\mathbf{G}$. We assume the instantaneous eigenstates split into two groups of bands which are separated by a large bandgap $\Delta$. In analogy with the main text we define the two groups as $L$-and $R$-bands, respectively, i.e. $\tilde{\psi} = (\tilde{R}, \tilde{L})$. The unitary transformation for the single-particle operators in this notation is

$$\begin{aligned}
\tilde{R}_\rho^\dagger(\mathbf{k}, t) &= \tilde{\alpha}_{\rho,\nu}(\mathbf{k}, t)A_\nu^\dagger(\mathbf{k}) &&\text{for } \rho = 1, ..., N_R; \\
\tilde{L}_\lambda^\dagger(\mathbf{k}, t) &= \tilde{\alpha}_{\lambda,\nu}(\mathbf{k}, t)A_\nu^\dagger(\mathbf{k}) &&\text{for } \lambda = N_R + 1, ..., N.
\end{aligned} \tag{99}$$

In the following the indices $\nu$ and $\tau$ always run over all bands, while $\rho$ indexes the $R$- and $\lambda$ the $L$-bands only, respectively. A generic two-particle interaction between all bands can be written as

$$\mathcal{V} = U \sum_{\mathbf{x}, \{\mathbf{d}_i\}} \chi_{\nu_1,\nu_2}^{\nu_3,\nu_4}(\{\mathbf{d}_i\})A_{\nu_1}^\dagger(\mathbf{x} + \mathbf{d}_1)A_{\nu_2}^\dagger(\mathbf{x} + \mathbf{d}_2)A_{\nu_3}(\mathbf{x} + \mathbf{d}_3)A_{\nu_4}(\mathbf{x}). \tag{100}$$

We assume a quasi-local interaction which means that $|\chi_{\nu_1,\nu_2}^{\nu_3,\nu_4}(\{\mathbf{d}_i\})|$ decays exponentially at large $\|\mathbf{d}_i\|$, and we choose $\chi$ and $U$ so that $\sum_{\{\mathbf{d}_i\},\{\nu_i\}} |\chi_{\nu_1,\nu_2}^{\nu_3,\nu_4}(\{\mathbf{d}_i\})| = 1$. The intraband interaction does not change the population of the $R$-bands, but as in the main text it splits into parts which act on two particles on the $R$-bands, two particles in the $L$-bands, or one particle in each

group of bands. In momentum space the *RR*-term is

$$
\tilde{\mathcal{V}}_{intra}^{RR}(t) = \frac{U}{V^2} \sum_{\mathbf{x},\{\mathbf{d}_i\},\{\mathbf{k}_i\}} \chi_{v_1,v_2}^{v_3,v_4}(\{\mathbf{d}_i\}) e^{i(\mathbf{k}_1(\mathbf{x}+\mathbf{d}_1)+\mathbf{k}_2(\mathbf{x}+\mathbf{d}_2)-\mathbf{k}_3(\mathbf{x}+\mathbf{d}_3)-\mathbf{k}_4\mathbf{x})}
$$

$$
\tilde{\alpha}_{\rho_1,v_1}^*(\mathbf{k}_1,t)\tilde{\alpha}_{\rho_2,v_2}^*(\mathbf{k}_2,t)\tilde{\alpha}_{\rho_3,v_3}(\mathbf{k}_3,t)\tilde{\alpha}_{\rho_4,v_4}(\mathbf{k}_4,t)
$$

$$
\tilde{R}_{\rho_1}^\dagger(\mathbf{k}_1,t)\tilde{R}_{\rho_2}^\dagger(\mathbf{k}_2,t)\tilde{R}_{\rho_3}(\mathbf{k}_3,t)\tilde{R}_{\rho_4}(\mathbf{k}_4,t)
$$

$$
\equiv \frac{U}{V^2} \sum_{\mathbf{x},\{\mathbf{k}_i\}} e^{i\mathbf{x}(\mathbf{k}_1+\mathbf{k}_2-\mathbf{k}_3-\mathbf{k}_4)}(\tilde{g}_{RR})_{\rho_1,\rho_2}^{\rho_3,\rho_4}(\{\mathbf{k}_i\},t)\tilde{R}_{\rho_1}^\dagger(\mathbf{k}_1)\tilde{R}_{\rho_2}^\dagger(\mathbf{k}_2)\tilde{R}_{\rho_3}(\mathbf{k}_3)\tilde{R}_{\rho_4}(\mathbf{k}_4), \tag{101}
$$

with the implicit definition of $(\tilde{g}_{RR})_{\rho_1,\rho_2}^{\rho_3,\rho_4}(\{\mathbf{k}_i\},t)$. Similarly we obtain the *LL*- and the *LR*-terms where the latter has a prefactor $\tilde{g}_{LR}$ which consists of 4 terms, as in Eq. (14) in the main text. $V$ is the ($d$-dimensional) volume of the system.

The interband part of the interaction which excites one particle from the $R$- to the $L$-bands splits into two pieces, one part $\tilde{\mathcal{V}}_+^{RR\to RL}(t)$ that acts on two particles in the $R$-bands and one part $\tilde{\mathcal{V}}_+^{RL\to LL}(t)$ that acts on one particle in the $R$- and $L$-bands each. The first part becomes

$$
\tilde{\mathcal{V}}_+^{RR\to RL}(t) = \frac{U}{V^2} \sum_{\mathbf{x},\{\mathbf{d}_i\},\{\mathbf{k}_i\}} \chi_{v_1,v_2}^{v_3,v_4}(\{\mathbf{d}_i\}) e^{i(\mathbf{k}_1(\mathbf{x}+\mathbf{d}_1)+\mathbf{k}_2(\mathbf{x}+\mathbf{d}_2)-\mathbf{k}_3(\mathbf{x}+\mathbf{d}_3)-\mathbf{k}_4\mathbf{x})}
$$

$$
\left( \tilde{\alpha}_{\lambda_1,v_1}^*(\mathbf{k}_1,t)\tilde{\alpha}_{\rho_2,v_2}^*(\mathbf{k}_2,t) - \tilde{\alpha}_{\lambda_1,v_2}^*(\mathbf{k}_1,t)\tilde{\alpha}_{\rho_2,v_1}^*(\mathbf{k}_2,t) \right)
$$

$$
\tilde{\alpha}_{\rho_3,v_3}(\mathbf{k}_3,t)\tilde{\alpha}_{\rho_4,v_4}(\mathbf{k}_4,t)\tilde{L}_{\lambda_1}^\dagger(\mathbf{k}_1,t)\tilde{R}_{\rho_2}^\dagger(\mathbf{k}_2,t)\tilde{R}_{\rho_3}(\mathbf{k}_3,t)\tilde{R}_{\rho_4}(\mathbf{k}_4,t)
$$

$$
\equiv \frac{U}{V^2} \sum_{\mathbf{x},\{\mathbf{k}_i\}} e^{i\mathbf{x}(\mathbf{k}_1+\mathbf{k}_2-\mathbf{k}_3-\mathbf{k}_4)}(\tilde{f}_R)_{\lambda_1,\rho_2}^{\rho_3,\rho_4}(\{\mathbf{k}_i\},t)\tilde{L}_{\lambda_1}^\dagger(\mathbf{k}_1)\tilde{R}_{\rho_2}^\dagger(\mathbf{k}_2)\tilde{R}_{\rho_3}(\mathbf{k}_3)\tilde{R}_{\rho_4}(\mathbf{k}_4) \tag{102}
$$

$$
\equiv \sum_{\mathbf{x}} \tilde{\mathcal{V}}_{\mathbf{x}}^{RR\to RL}(t),
$$

and similarly for $\tilde{\mathcal{V}}_+^{RL\to LL}(t)$. We want to calculate the rate at which this interband interaction changes the particle density in the $L$-bands $n_L = N_L/V$:

$$
\Gamma = \frac{1}{V\mathcal{T}} \sum_f \iint_0^{\mathcal{T}} dt' dt \sum_{\mathbf{x}',\mathbf{x}} \langle \tilde{\Psi}_i(t')|\tilde{\mathcal{V}}_{\mathbf{x}'}^\dagger(t')|\tilde{\Psi}_f(t')\rangle\langle\tilde{\Psi}_f(t)|\tilde{\mathcal{V}}_{\mathbf{x}}(t)|\tilde{\Psi}_i(t)\rangle. \tag{103}
$$

We start by separating nearby and distant terms as in section 4.1, determined by a critical distance $r^*$:

$$
\sum_{\mathbf{x},\mathbf{x}'} = \sum_{\mathbf{x}} \sum_{\|\mathbf{x}'-\mathbf{x}\|>r^*} + \sum_{\mathbf{x}} \sum_{\|\mathbf{x}'-\mathbf{x}\|\leq r^*}. \tag{104}
$$

After changing to the Heisenberg picture and completing the commutator as in Eq. (17) we apply the Lieb-Robinson bounds and obtain

$$
\Gamma_{dist}(\mathcal{T}) = \frac{1}{V\mathcal{T}} \iint_0^{\mathcal{T}} dt' dt \sum_{\mathbf{x}} \sum_{|\mathbf{x}'-\mathbf{x}|>r^*} \langle\tilde{\Psi}_i|[\tilde{\mathcal{V}}_{\mathbf{x}'}^{H\dagger}(t'),\tilde{\mathcal{V}}_{\mathbf{x}}^H(t)]|\tilde{\Psi}_i\rangle \tag{105}
$$

$$
\leq \frac{2CS_d}{c^3 v^2}\frac{U^2}{\mathcal{T}}e^{-cr^*}\left(e^{cv\mathcal{T}}-1-cv\mathcal{T}\right).
$$

Here $S_d$ is the surface of the $d$-dimensional unit sphere, i.e. $S_1 = 2, S_2 = 2\pi, S_3 = 4\pi$. It is worth noting that the Lieb-Robinson velocity $v$ also depends on the localization length of the intraband interaction $\tilde{\mathcal{V}}_{\mathbf{x}}(t)$ which includes the localization length of $\chi_{v_1,v_2}^{v_3,v_4}(\{\mathbf{d}_i\})$ in (100).

Turning to the nearby terms we apply a rotating frame transformation akin to section 4.2.1:

$$
R_\rho^\dagger(\mathbf{k},t) = \tilde{R}_\rho^\dagger(\mathbf{k},t)e^{-i\Delta t/2} \quad ; \quad L_\lambda^\dagger(\mathbf{k},t) = \tilde{L}_\lambda^\dagger(\mathbf{k},t)e^{i\Delta t/2}. \tag{106}
$$

This implies for the entries in the unitary matrix $\alpha$ that

$$\begin{aligned}
\alpha_{\rho,\nu}(\mathbf{k},t) &= \tilde{\alpha}_{\rho,\nu}(\mathbf{k},t)e^{-i\Delta t/2} && \text{for } \rho = 1,...,N_R; \\
\alpha_{\lambda,\nu}(\mathbf{k},t) &= \tilde{\alpha}_{\lambda,\nu}(\mathbf{k},t)e^{i\Delta t/2} && \text{for } \lambda = N_R + 1,...,N.
\end{aligned} \tag{107}$$

In the rotated Hamiltonian

$$\mathcal{H}_0(t) = \sum_{\mathbf{k}} \psi_\nu^\dagger(\mathbf{k},t)E_{\nu,\tau}(\mathbf{k},t)\psi_\tau(\mathbf{k},t) \tag{108}$$

all matrix elements in the diagonal blocks shift to a narrow interval of width $W$ (cf. Eq. (24)),

$$\left|E_{\rho,\rho'}(\mathbf{k},t), E_{\lambda,\lambda'}(\mathbf{k},t)\right| \le \frac{W}{2}, \quad \text{for } \rho,\rho' = 1,...,N_R;\ \lambda,\lambda' = N_R + 1,...,N. \tag{109}$$

In analogy with Eq. (25) we define the dimensionless quantities $e_{\nu,\tau}(\mathbf{k},t) \equiv E_{\nu,\tau}(\mathbf{k},t)/W$ to write the Hamiltonian as

$$\mathcal{H}_0(t) = W \sum_{\mathbf{k}} \psi_\nu^\dagger(\mathbf{k},t)e_{\nu,\tau}(\mathbf{k},t)\psi_\tau(\mathbf{k},t). \tag{110}$$

After applying a triangle inequality similar to Eq. (22) the excitation probability in the rotated frame becomes

$$\begin{aligned}
\Gamma_{near}(\mathcal{T}) \ &\le\ \frac{V^*}{V\mathcal{T}} \sum_f \sum_{\mathbf{x}} \left|\int_0^{\mathcal{T}} dt \left\langle \Psi_f(t)\right| \mathcal{V}_{\mathbf{x}}(t) \left|\Psi_i(t)\right\rangle e^{i\Delta t}\right|^2 \\
&\le\ \frac{2V^*}{V\mathcal{T}} \sum_f \sum_{\mathbf{x}} \left|\left[\sum_{m=0}^{M-1} \frac{-i}{\Delta} e^{i\Delta t} \left(\frac{i\partial_t}{\Delta}\right)^m \left\langle \Psi_f(t)\right| \mathcal{V}_{\mathbf{x}}(t) \left|\Psi_i(t)\right\rangle\right]_{t=0}^{\mathcal{T}}\right|^2 \\
&+\ \frac{2V^*}{V\mathcal{T}} \sum_f \sum_{\mathbf{x}} \left|\int_0^{\mathcal{T}} dt\, e^{i\Delta t} \left(\frac{i\partial_t}{\Delta}\right)^M \left\langle \Psi_f(t)\right| \mathcal{V}_{\mathbf{x}}(t) \left|\Psi_i(t)\right\rangle\right|.
\end{aligned} \tag{111}$$

Here $V^*$ is the volume of a ball of radius $r^*$, and in the second step we introduced derivatives $e^{i\Delta t} = (-i\partial_t)^M \Delta^{-M} e^{i\Delta t}$ and performed integration by parts. From there we derive a similar result for the excitation probability as in Eq. (37),

$$\Gamma_{near} \le \frac{8\pi V^*}{\omega} \|\mathcal{O}_{\mathbf{x}}^M\|^2 + \frac{8V^*}{\mathcal{T}} \left(\frac{U}{\Delta}\right)^2, \tag{112}$$

in which the rate of continued excitations only depends on the norm of the operator $\|\mathcal{O}_{\mathbf{x}}^M(t)\|$ which is implicitly defined by

$$\left\langle \Psi_f(t)\right| \mathcal{O}_{\mathbf{x}}^M(t) \left|\Psi_i(t)\right\rangle \equiv \left(\frac{i\partial_t}{\Delta}\right)^M \left\langle \Psi_f(t)\right| \mathcal{V}_{\mathbf{x}}(t) \left|\Psi_i(t)\right\rangle. \tag{113}$$

Note that, as Eq. (35) in the main text, Eq. (113) does not uniquely define the operator $\|\mathcal{O}_{\mathbf{x}}^M(t)\|$, but we only need to consider one operator which fulfills the condition (113).

To find a bound on the norm of this operator we follow the same steps as in section 4.2.3. We consider iterative application of the $M$-fold derivative, where in each step the effect is given by the Heisenberg equation (39),

$$\frac{i\partial_t}{\Delta} \left\langle \Psi_f(t)\right| \mathcal{V}_{\mathbf{x}}(t) \left|\Psi_i(t)\right\rangle = \left\langle \Psi_f(t)\right| \left[\frac{1}{\Delta}\mathcal{V}_{\mathbf{x}}(t), \mathcal{H}_0(t) + \mathcal{V}_{intra}(t)\right] + \left(\frac{i\partial_t}{\Delta}\mathcal{V}_{\mathbf{x}}(t)\right) \left|\Psi_i(t)\right\rangle. \tag{114}$$

From now on we only consider the operator $\mathcal{V}_{\mathbf{x}}^{\text{RR}\to\text{RL}}(t)$, the contribution from $\mathcal{V}_{\mathbf{x}}^{\text{RL}\to\text{LL}}(t)$ is calculated analogously.

First we consider the commutator of $\mathcal{V}_{\mathbf{x}}^{\text{RR}\to\text{RL}}(t)$ with the quadratic Hamiltonian. The elements in the off-diagonal blocks change the occupation of the $L$-and $R$-band, therefore

$$\left\langle \Psi_f \left| \left[ L_{\lambda_1}^\dagger(\mathbf{k}_1) R_{\rho_2}^\dagger(\mathbf{k}_2) R_{\rho_3}(\mathbf{k}_3) R_{\rho_4}(\mathbf{k}_4), \sum_{\mathbf{k}} (e_{\rho,\lambda}(\mathbf{k}) R_\rho^\dagger(\mathbf{k}) L_\lambda(\mathbf{k}) + h.c.) \right] \right| \Psi_i \right\rangle = 0. \qquad (115)$$

Hence we only need to account for the block on the diagonal of $\mathcal{H}_0$ in (24) with $R^\dagger R$ and $L^\dagger L$. The commutator gives

$$\left[ \frac{1}{\Delta} \mathcal{V}_{\mathbf{x}}^{\text{RR}\to\text{RL}}(t), \sum_{\mathbf{k}} e_{\rho,\rho'}(\mathbf{k}) R_\rho^\dagger(\mathbf{k}) R_{\rho'}(\mathbf{k}) + e_{\lambda,\lambda'}(\mathbf{k}) L_\lambda^\dagger(\mathbf{k}) L_{\lambda'}(\mathbf{k}) \right] = \qquad (116)$$

$$= \frac{U}{\Delta} \frac{1}{V^2} \sum_{\{\mathbf{k}_i\}} e^{i\mathbf{x}(\mathbf{k}_1+\mathbf{k}_2-\mathbf{k}_3-\mathbf{k}_4)} L_{\lambda_1}^\dagger(\mathbf{k}_1) R_{\rho_2}^\dagger(\mathbf{k}_2) R_{\rho_3}(\mathbf{k}_3) R_{\rho_4}(\mathbf{k}_4)$$

$$\sum_{\{\mathbf{d}_i\}} \left\{ -(f_{\text{R}})_{\lambda,\rho_2}^{\rho_3,\rho_4}(\{\mathbf{k}_i\},t) e_{\lambda_1,\lambda}(\mathbf{k}_1,t) - (f_{\text{R}})_{\lambda_1,\rho}^{\rho_3,\rho_4}(\{\mathbf{k}_i\},t) e_{\rho_2,\rho}(\mathbf{k}_2,t) \right.$$

$$\left. + (f_{\text{R}})_{\lambda_1,\rho_2}^{\rho,\rho_4}(\{\mathbf{k}_i\},t) e_{\rho,\rho_3}(\mathbf{k}_3,t) + (f_{\text{R}})_{\lambda_1,\rho_2}^{\rho_3,\rho}(\{\mathbf{k}_i\},t) e_{\rho,\rho_4}(\mathbf{k}_4,t) \right\}.$$

As in the case with two bands this gives 4 new terms, however each of them contains a sum over all $L$- or all $R$-bands.

Next we calculate the commutator of the interaction with the intraband operator $\mathcal{V}_{intra}(t)$. For example, in the multi-band system the commutator with the $RR$-part $\mathcal{V}_{intra}^{\text{RR}}(t)$ takes the form

$$\left[ \frac{1}{\Delta} \mathcal{V}_{\mathbf{x}}^{\text{RR}\to\text{RL}}, \mathcal{V}_{intra}^{\text{RR}} \right] = \frac{U^2}{\Delta V^4} \sum_{\{\mathbf{k}_i\},\{\mathbf{q}_i\}} e^{i\mathbf{x}(\mathbf{k}_1+\mathbf{k}_2-\mathbf{k}_3-\mathbf{k}_4)} e^{i\mathbf{x}'(\mathbf{q}_1+\mathbf{q}_2-\mathbf{q}_3-\mathbf{q}_4)} \qquad (117)$$

$$(f_{\text{R}})_{\lambda_1,\rho_2}^{\rho_3,\rho_4}(\{\mathbf{k}_i\})(g_{RR})_{\gamma_1,\gamma_2}^{\gamma_3,\gamma_4}(\{\mathbf{q}_i\})$$

$$\left[ L_{\lambda_1}^\dagger(\mathbf{k}_1) R_{\rho_2}^\dagger(\mathbf{k}_2) R_{\rho_3}(\mathbf{k}_3) R_{\rho_4}(\mathbf{k}_4), R_{\gamma_1}^\dagger(\mathbf{q}_1) R_{\gamma_2}^\dagger(\mathbf{q}_2) R_{\gamma_3}(\mathbf{q}_3) R_{\gamma_4}(\mathbf{q}_4) \right],$$

where the $\gamma_i$ index the $R$-bands only. As in the main text, the commutator with the full intraband operator gives 24 terms of the form

$$\frac{U^2}{\Delta V^3} \sum_{\{\mathbf{k}_i\}} e^{i\mathbf{x}(\mathbf{k}_1+\mathbf{k}_2+\mathbf{k}_3-\mathbf{k}_4-\mathbf{k}_5-\mathbf{k}_6)} h_{\lambda_1,\rho_2,\rho_3}^{\rho_4,\rho_5,\rho_6}(\{\mathbf{k}_i\}) \times$$

$$L_{\lambda_1}^\dagger(\mathbf{k}_1) R_{\rho_2}^\dagger(\mathbf{k}_2) R_{\rho_3}^\dagger(\mathbf{k}_3) R_{\rho_4}(\mathbf{k}_4) R_{\rho_5}(\mathbf{k}_5) R_{\rho_6}(\mathbf{k}_6), \quad (118)$$

where the prefactor function for one example term is

$$h_{\lambda_1,\rho_2,\rho_3}^{\rho_4,\rho_5,\rho_6}(\{\mathbf{k}_i\}) = \sum_{\{\mathbf{d}_i,\mathbf{d}_i'\}} \chi_{\nu_1,\nu_2}^{\nu_3,\nu_4}(\{\mathbf{d}_i\}) \chi_{\tau_1,\tau_2}^{\tau_3,\tau_4}(\{\mathbf{d}_i\}) \qquad (119)$$

$$e^{i(\mathbf{k}_1\mathbf{d}_1+\mathbf{k}_2\mathbf{d}_2+\mathbf{k}_3(\mathbf{d}_1'-\mathbf{d}_2')-\mathbf{k}_4\mathbf{d}_3-\mathbf{k}_5(\mathbf{d}_3'-\mathbf{d}_2')+\mathbf{k}_6(\mathbf{d}_2-\mathbf{d}_2'))}$$

$$\left[ \alpha_{\lambda_1,\nu_1}^*(\mathbf{k}_1) \alpha_{\rho_2,\nu_2}^*(\mathbf{k}_2) - \alpha_{\lambda_1,\nu_2}^*(\mathbf{k}_1) \alpha_{\rho_2,\nu_1}^*(\mathbf{k}_2) \right] \alpha_{\rho_3,\tau_1}^*(\mathbf{k}_3) \alpha_{\rho_4,\nu_3}(\mathbf{k}_4)$$

$$\alpha_{\rho,\tau_2}^*(\mathbf{k}_5+\mathbf{k}_6-\mathbf{k}_3) \alpha_{\rho,\nu_4}(\mathbf{k}_5+\mathbf{k}_6-\mathbf{k}_3) \alpha_{\rho_5,\tau_3}(\mathbf{k}_5) \alpha_{\rho_6,\tau_4}(\mathbf{k}_6).$$

To calculate the explicit time derivative of $\mathcal{V}_{\mathbf{x}}^{\text{RR}\to\text{RL}}(t)$ we separate the time-dependent terms by their momenta $\mathbf{k}_i$, as in Eq. (67). From Eq. (98) we see that the functions $\alpha_{\nu,\tau}(\mathbf{k},t)$ are not

periodic in time, however as in the case of two bands in Eq. (68) the relevant products are. We calculate in a similar manner:

$$
\begin{aligned}
\frac{i\partial_t}{\Delta}\alpha_{\rho_4,\nu_4}(\mathbf{k}_4,t)R_{\rho_4}(\mathbf{k}_4,t) &= \frac{i\partial_t}{\Delta}\alpha_{\rho_4,\nu_4}(\mathbf{k}_4,t)\alpha^*_{\rho_4,\nu}(\mathbf{k}_4,t)A_\nu(\mathbf{k}_4) \\
&= \frac{\omega}{\Delta}\sum_l le^{-il\omega t}\left(\alpha_{\rho_4,\nu_4}(\mathbf{k}_4)\alpha^*_{\rho_4,\nu}(\mathbf{k}_4)\right)^{(l)}A_\nu(\mathbf{k}_4) \\
&= \frac{\omega}{\Delta}\sum_l le^{-il\omega t}\left(\alpha_{\rho_4,\nu_4}(\mathbf{k}_4)\alpha^*_{\rho_4,\nu}(\mathbf{k}_4)\right)^{(l)} \\
&\qquad\left(\alpha_{\rho,\nu}(\mathbf{k}_4,t)R_\rho(\mathbf{k}_4,t)+\alpha_{\lambda,\nu}(\mathbf{k}_4,t)L_\lambda(\mathbf{k}_4,t)\right).
\end{aligned}
\tag{120}
$$

The part with the operator $L_\lambda(\mathbf{k}_4,t)$ reduces the number of particles in the $L$-band, therefore it does not connect to the final state in the matrix. We can drop this part and only keep the operator $R_\rho(\mathbf{k}_4,t)$. Performing this calculation for all four momenta $\mathbf{k}_i$ gives four new terms of the form

$$
\frac{\omega}{\Delta}\sum_{\{\mathbf{k}_i\}}e^{i\mathbf{x}(\mathbf{k}_1+\mathbf{k}_2-\mathbf{k}_3-\mathbf{k}_4)}G^{\rho_3,\rho_4}_{\lambda_1,\rho_2}(\{\mathbf{k}_i\},t)L^\dagger_{\lambda_1}(\mathbf{k}_1,t)R^\dagger_{\rho_2}(\mathbf{k}_2,t)R_{\rho_3}(\mathbf{k}_3,t)R_{\rho_4}(\mathbf{k}_4,t),
\tag{121}
$$

where in the example from Eq. (120) the resulting prefactor function is

$$
\begin{aligned}
G^{\rho_3,\rho_4}_{\lambda_1,\rho_2}(\{\mathbf{k}_i\},t) &= \sum_{\{\mathbf{d}_i\}}e^{i(\mathbf{k}_1\mathbf{d}_1+\mathbf{k}_2\mathbf{d}_2-\mathbf{k}_3\mathbf{d}_3-\mathbf{k}_4\mathbf{d}_4)}\chi^{\nu_3,\nu_4}_{\nu_1,\nu_2}(\{\mathbf{d}_i\}) \\
&\quad\left(\alpha^*_{\lambda_1,\nu_1}(\mathbf{k}_1,t)\alpha^*_{\rho_2,\nu_2}(\mathbf{k}_2,t)-\alpha^*_{\lambda_1,\nu_2}(\mathbf{k}_1,t)\alpha^*_{\rho_2,\nu_1}(\mathbf{k}_2,t)\right)\alpha_{\rho_3,\nu_3}(\mathbf{k}_3,t) \\
&\quad\sum_l le^{-il\omega t}\left(\alpha_{\rho_4,\nu_4}(\mathbf{k}_4)\alpha^*_{\rho_4,\nu}(\mathbf{k}_4)\right)^{(l)}\alpha_{\rho,\nu}(\mathbf{k}_4,t).
\end{aligned}
\tag{122}
$$

From this discussion we see that each individual step gives several terms which have the same shape as the initial term $\mathcal{V}_{\mathbf{x}}(t)$. Therefore we can easily iterate taking derivatives. Furthermore the number of terms that are generated does not change compared to the special case in the main text. Thus the total number $\mathcal{N}$ of final terms after taking $M$ derivatives is identical to the result in Eq. (46) in the main text, including the terms from $\mathcal{V}^{\mathrm{RR}\to\mathrm{RL}}_{\mathbf{x}}(t)$ and $\mathcal{V}^{\mathrm{RL}\to\mathrm{LL}}_{\mathbf{x}}(t)$ gives

$$
\mathcal{N}\le 5(16M)^M.
\tag{123}
$$

note that here each term contains sums over all bands, however this does not affect the rest of the calculation.

Similarly to Eq. (47) in the main text we can write the operator $\mathcal{O}^M_{\mathbf{x}}(t)$ in Eq. (113) as

$$
\begin{aligned}
\mathcal{O}^M_{\mathbf{x}}(t) &= \sum_{\eta=1}^{\mathcal{N}}U\frac{U^qW^p\omega^{M-p-q}}{\Delta^M}\frac{1}{V^{n/2}}\sum_{\{\mathbf{k}_i\}}e^{i\mathbf{x}(\mathbf{k}_1+\dots+\mathbf{k}_{n/2}-\mathbf{k}_{n/2+1}-\dots-\mathbf{k}_n)} \\
&\quad \mathcal{F}_{\eta,\lambda_1,\{\rho_i\}}(\{\mathbf{k}_i\},t)L^\dagger_{\lambda_1}(\mathbf{k}_1)R^\dagger_{\rho_2}(\mathbf{k}_2)\dots R^\dagger_{\rho_{n/2}}(\mathbf{k}_{n/2})R_{\rho_{n/2+1}}(\mathbf{k}_{n/2+1})\dots R_{\rho_n}(\mathbf{k}_n).
\end{aligned}
\tag{124}
$$

We rewrite the operators $R_\rho(\mathbf{k},t)$, $L_\lambda(\mathbf{k},t)$ in the sublattice basis according to Eq. (99). The functions $\alpha_{\nu,\nu'}(\mathbf{k})$ and the functions $\mathcal{F}_{\eta,\lambda_1,\{\rho_i\}}(\{\mathbf{k}_i\},t)$ are periodic in all momenta $\mathbf{k}_i$. We write them in their Fourier series akin to equations (75) and (74). Summing over all momenta and using $\|A_\nu(\mathbf{x})\|=1$ for the norm of the on-site operators we find

$$
\begin{aligned}
\|\mathcal{O}^M_{\mathbf{x}}(t)\| &= \sum_{\eta=1}^{\mathcal{N}}U\frac{U^qW^p\omega^{M-p-q}}{\Delta^M}\sum_{\{\mathbf{y}_i,\mathbf{z}_i\},\{\nu_i\}} \\
&\quad\left|\sum_{\lambda_1,\{\rho_i\}}\overline{\mathcal{F}}_{\eta,\lambda_1,\{\rho_i\}}(\{\mathbf{z}_i\})\overline{\alpha}_{\lambda_1,\nu_1}(\mathbf{y}_1)\dots\overline{\alpha}_{\rho_{n/2},\nu_{n/2}}(\mathbf{y}_{n/2})\overline{\alpha}^*_{\rho_{n/2+1},\nu_{n/2+1}}(\mathbf{y}_{n/2+1})\dots\overline{\alpha}^*_{\rho_n,\nu_n}(\mathbf{y}_n)\right|.
\end{aligned}
\tag{125}
$$

The absolute values of $\overline{\mathcal{F}}$ and $\overline{\alpha}$ decay exponentially at large $\|\mathbf{y}_i, \mathbf{z}_i\|$ because of the $\mathbf{k}$-periodicity of $\mathcal{F}$ and $\alpha$. Thus the sums over $\mathbf{y}_i, \mathbf{z}_i$ in Eq. (125) converge to a finite value. In the initial term with $M = 0$ derivatives we have $\mathcal{F}_1 = f_R$ and $\mathcal{F}_2 = f_L$. We bound the constant by

$$\sum_{\{\mathbf{y}_i, \mathbf{z}_i\}, \{v_i\}} \left| \sum_{\lambda_1, \{\rho_i\}} (\overline{f}_R)_{\lambda_1, \rho_2}^{\rho_3, \rho_4}(\{\mathbf{z}_i\}) \overline{\alpha}_{\lambda_1, v_1}(\mathbf{y}_1) \overline{\alpha}_{\rho_2, v_2}(\mathbf{y}_{n/2}) \overline{\alpha}_{\rho_3, v_3}^*(\mathbf{y}_3) \overline{\alpha}_{\rho_4, v_4}^*(\mathbf{y}_4) \right|$$

$$\leq \sum_{\{\mathbf{y}_i, \mathbf{z}_i\}, \{\mathbf{d}_i\}, \{v_i\}} \left| \chi_{\tau_1, \tau_2}^{\tau_3, \tau_4}(\{\mathbf{d}_i\}) \overline{\alpha}_{\lambda_1, v_1}(\mathbf{y}_1) \overline{\alpha}_{\rho_2, v_2}(\mathbf{y}_2) \overline{\alpha}_{\rho_3, v_3}^*(\mathbf{y}_3) \overline{\alpha}_{\rho_4, v_4}^*(\mathbf{y}_4) \right.$$

$$\left. \left( \overline{\alpha}_{\lambda_1, \tau_1}^*(\mathbf{z}_1) \overline{\alpha}_{\rho_2, \tau_2}^*(\mathbf{z}_2) - \overline{\alpha}_{\lambda_1, \tau_2}^*(\mathbf{z}_1) \overline{\alpha}_{\rho_2, \tau_1}^*(\mathbf{z}_2) \right) \overline{\alpha}_{\rho_3, \tau_3}(\mathbf{z}_3) \overline{\alpha}_{\rho_4, \tau_4}(\mathbf{z}_4) \right| \equiv K. \quad (126)$$

As in appendix C.3 we now consider the changes to this constant $K$ that come from taking one commutator or one derivative. In all cases the norm of $\mathcal{F}$ only obtains an additional factor. The commutator with the quadratic Hamiltonian $\mathcal{H}_0$ changes

$$\sum_{\mathbf{z}_4} |\overline{\alpha}_{\rho_4, \tau_4}(\mathbf{z}_4)| \to \frac{1}{W} \sum_{\mathbf{z}_4, \mathbf{w}} |\overline{\alpha}_{\rho, \tau_4}(\mathbf{z}_4) \overline{e}_{\rho, \rho_4}(\mathbf{w} - \mathbf{z}_4)| \quad (127)$$

in the expression of $|\overline{\mathcal{F}}_\eta|$. We define the constant $K_E$ analog to the definition in Eq. (82) so that

$$\sum_{\{\mathbf{y}_i, \mathbf{z}_i\}, \{v_i\}} \left| \overline{\mathcal{F}}_{\eta, \lambda_1, \{\rho_i\}}(\{\mathbf{z}_i\}) \overline{\alpha}_{\lambda_1, v_1}(\mathbf{y}_1) ... \overline{\alpha}_{\rho_n, v_n}^*(\mathbf{y}_n) \right| \leq K_E K. \quad (128)$$

The change to Eq. (126) due to the commutator with the intraband interaction $\mathcal{V}_{intra}$ is

$$\sum_{\mathbf{z}_4, \rho_4, v_4} |\overline{\alpha}_{\rho_4, v_4}(\mathbf{z}_4)| \to \sum_{\mathbf{w}_{1,2}, \mathbf{z}_{4,5,6}, \rho_{4,5,6}, v_4, \{\mathbf{d}_i\}}$$

$$\left| \chi_{\gamma_1, \gamma_2}^{\gamma_3, \gamma_4}(\{\mathbf{d}_i\}) \overline{\alpha}_{\rho_4, \gamma_1}^*(\mathbf{z}_4) \overline{\alpha}_{\rho, \gamma_2}^*(\mathbf{w}_1) \overline{\alpha}_{\rho, v_4}(\mathbf{w}_2) \overline{\alpha}_{\rho_5, \gamma_3}(\mathbf{z}_5) \overline{\alpha}_{\rho_6, \gamma_4}(\mathbf{z}_6) \right|. \quad (129)$$

After the smoke clears this merely multiplies the constant $K$ by a numerical factor. We define this factor as $K_U$, in analogy with $K_E$ in Eq. (128). The explicit time-derivative of the operator yields the change

$$\sum_{\mathbf{z}_4, \rho_4, v_4} |\overline{\alpha}_{\rho_4, v_4}(\mathbf{z}_4)| \to \sum_{\mathbf{w}_{1,2}, \mathbf{z}_4, \rho_4, v_4} \left| \sum_l l e^{-il\omega t} \left( \overline{\alpha}_{\rho, v_4}(\mathbf{w}_1) \overline{\alpha}_{\rho, v}^*(\mathbf{w}_2) \right)^{(l)} \overline{\alpha}_{\rho_4, v}(\mathbf{z}_4) \right|. \quad (130)$$

We define the additional contribution to the constant in Eq. (126) as $K_\omega$. We want to stress again that even though these constants all contain sums over all coefficients in the Fourier series and over all bands, due to the periodicity of the functions the Fourier coefficients go to zero exponentially fast and the constants are all finite. We obtain the same result for the operator norm as in Eq. (50) in the main text,

$$\|\mathcal{O}_x^M(t)\| \leq 5KU \left( \frac{16M\mathcal{E}}{\Delta} \right)^M = 5KU e^{-\mu\Delta/\mathcal{E}}, \quad (131)$$

where the constant $K$ is defined in Eq. (126). In the last step we again choose $M = \frac{\mu\Delta}{\mathcal{E}}$ with $\mu = (16e)^{-1}$, and we use the renormalized energy scales $\mathcal{E} = \max\{K_U U, K_E W, K_\omega \omega\}$. This is identical to the previous result, only the constants $K_{U,E,\omega}$ that renormalize the energy scales change. Applying this to the excitation probability in Eq. (111) and adding the contribution from the distant terms in Eq. (105) gives

$$\Gamma \leq 200\pi K^2 V^* \frac{U^2}{\omega} e^{-2\mu\Delta/\mathcal{E}} + 8V^* \left( \frac{U}{\Delta\mathcal{T}} \right)^2 + \frac{2CS_d}{c^3 v^2} \frac{U^2}{\mathcal{T}} e^{-cr^*} \left( e^{cv\mathcal{T}} - 1 - cv\mathcal{T} \right). \quad (132)$$

This final result is the direct generalization for the special case of a two-band system in one dimension, Eq. (51). In any dimension the excitation rate grows linearly in time with a heating rate that is exponentially small in $\frac{\Delta}{\varepsilon}$. The rate of change of the particle densities in the two bands is independent of the system volume which is expected from the thermodynamic limit. The tighter bound for a system with initially empty $L$-bands in appendix D also generalizes straightforwardly to the multiband system.

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
