# Peer review of "Exponentially long lifetime of universal quasi-steady states in topological Floquet pumps"

_SciPost Physics, doi:SciPost Phys. 9, 015 (2020)_

## Round 2 · Referee Report · Anonymous (Referee 1) · 2019-12-28

Report

In the manuscript by Gulden et. al, the authors analytically proved the existence of a time window in which a prethermal quasi-steady state persists, given the following conditions: 1) Local Hamiltonians; 2) Single-particle energy gap is the largest energy scale.
I find this result interesting and important, and the paper is well written, except some minor things listed below. Once fixed, the paper can be published.

-- Page 5, second paragraph, i∂_t |ψ ̃ ⟩=H_0 (t)|ψ ̃ ⟩, H_0 (t) is not defined.
-- 2 lines below the above one, should the H_0 (t) have a tilde on top of it?
-- Page 6, preservesthe -> preserves the
-- Page 6, below Eq.(4), one should define L as the length of the system.
--Page 9, below Eq.(9), the authors wrote “similarly for β _k (t)”. How about θ _k (t)?
-- Page 15, in Eq.(39), the authors should define V_0 (t). Is it the same as V_intra?
-- Page 21, end of Appendix A, V(t) should be \mu(t)?

---

## Round 2 · Referee Report · Anonymous (Referee 2) · 2020-1-19

Strengths

  1. This paper addresses a topic that is timely and important. There are many proposals for realizing nontrivial topological states in Floquet systems at the level of band structure, but the stability of these in the presence of interactions is rather poorly studied. Thus the work is well-motivated.

  2. The technical arguments seem reasonable (and are analogous to other recent results on thermalization, so I am fairly confident the main steps in the proof are correct).

Weaknesses

  1. There is a point I still find confusing and did not see clearly addressed in the text. If the instantaneous bands are very far apart in energy, then their hybridization to form a Floquet band is also exponentially suppressed in the same parameter Delta. The concept of a Floquet band is only sensible on timescales long compared with hybridization. The authors do not dwell on this point but it seems crucial. (Absent hybridization one cannot get a nontrivial Floquet model.)

  2. Generally a little more discussion of the relevant perturbation theory would be helpful. I am not fully up to date on recent work on this topic, so I am probably missing something obvious, but since this paper is not page-limited it has no excuse for not being self-contained.

  3. For Floquet bands with very small band gaps one would worry that a small electric field (for example) would drive diabatic transitions between the right- and left- pumping bands.

Report

I assume the main result is right since I cannot find concrete errors in the proofs but I am missing part of the intuition for why. It seems to be using the instantaneous band gap to control heating rates, which is fine, but it seems that this will also lead to Floquet bands with very small Floquet band gaps.

I would like the authors to consider these issues when revising.
  • validity: high
  • significance: high
  • originality: good
  • clarity: good
  • formatting: perfect
  • grammar: perfect

Author:  Tobias Gulden  on 2020-02-20  [id 744]

(in reply to Report 2 on 2020-01-19)
Category:
question

Dear referee,
we thank you for your thorough and detailed analysis of our work! While we are preparing a response and corrections to our manuscript, we are not entirely sure which point you address with your second comment "a little more discussion of the relevant perturbation theory would be helpful". Can you please point out the relevant location(s) in our manuscript, e.g. section or equation numbers? That would be very helpful for us to address this weakness as precisely as possible.
Many thanks!

---

## Round 3 · Referee Report · Anonymous · 2020-7-8

Report

The authors have addressed my concerns and the paper can be published

---

## Round 3 · Author Response

Dear editor and referees,

We thank all of you for the meticulous and detailed analysis of our work. The referees raised a few valid questions which we address in this reply and clarify at the relevant points in the revised version of the manuscript.

Reply to the comments of referee 2:

(1) The hybridization of the instantaneous bands to form Floquet bands is actually not exponentially small - the mixing of the instantaneous bands is of the order of \omega divided by the instantaneous gap \Delta. The existence of quantized pumping relies on this fact: the induced current in a quantized pump must be proportional to \omega to yield a pumped charge over a full period that is independent of \omega. The hybridization gap that is exponentially small in \Delta/\omega is the avoided crossing gap between the two Floquet bands (the crossing of the red and blue lines in Fig. 2; the gaps are too small to be visible in the figure). This is discussed in depth in Ref. [57] and supported by Ref. [68]. To define the right- and left- moving Floquet bands, these small gaps must be ignored. As the referee states, these gaps are only relevant at exponentially long time scales.

We added a clarification of this point to the discussion of the Floquet spectrum in Fig. 2.

(2) We thank the referee for asking for clarification.The perturbation theory we are using throughout the paper to estimate the inter-band scattering rate is just the standard Fermi's golden rule obtained from time-dependent perturbation theory (Eq. 4). The introduction of M time derivatives and factors of \Delta^{-1} [for example in Eq. (32)] is just a mathematical tool to bound the matrix element appearing in Fermi’s golden rule, with the value of M chosen such that the resulting bound is as tight as possible.

We have added a clarification of this point in the text, both before Eq. (32) and after Eq. (51).

(3) As explained in our answer to point 1, in order to define the right and left moving band it is necessary to neglect the exponentially small avoided-crossing gaps between the Floquet bands. This is shown in Figure 2, where the red color indicates the right moving band, and the blue color indicates the left moving band. If a small electric field is applied, there is an exponentially small probability to make a transition between right and left moving Floquet bands, which by Landau-Zener theory scales as (\Delta_ac)^2/E, where \Delta_ac is the exponentially small avoided-crossing gap, and E is the strength of the electric field. Correspondingly, with a probability which is exponentially close to 1, the particle would keep its identity as a right or a left mover as it sweeps through the crossing.

We have added a clarification in which we explain the avoided crossing between the Floquet bands in the first paragraph of sub-section 2.2.

Reply to the comments of referee 1:

We thank the referee for the attentive reading of our manuscript. We corrected the noted typos in the final version. Regarding \theta_k(t): its periodicity condition can be easily derived in the same way as the conditions for \alpha and \beta, $\theta_k(t+T) = \theta_k(t) + (\epsilon_{L,k}+\epsilon_{R,k})T mod 2\pi$. This condition is not relevant for the rest of the paper, therefore we opted not to include it.

We hope this reply and associated changes in the text have addressed all comments and questions raised by the two referees. We therefore believe that the updated manuscript should now be suitable for publication in SciPost, and look forward to its further processing.

With kind regards,

Tobias Gulden, Erez Berg, Mark Rudner, Netanel Lindner

---

## Round 3 · List of Changes

- The changes are noted in the author comments -

---

## Editorial Decision

published